# CryoEM structures of the human CLC-2 voltage-gated chloride channel reveal a ball-and-chain gating mechanism

**Mengyuan Xu[1], Torben Neelands[1], Alexander S Powers[2,3,4,5], Yan Liu[6], Steven D Miller[2], Grigore D Pintilie[7], J Du Bois[2], Ron O Dror[1,3,4,5], Wah Chiu[6,7], Merritt Maduke[1]***

[1]Department of Molecular and Cellular Physiology, Stanford University, Stanford, United States; [2]Department of Chemistry, Stanford University, Stanford, United States; [3]Department of Computer Science, Stanford University, Stanford, United States; [4]Department of Structural Biology, Stanford University, Stanford, United States; [5]Institute for Computational and Mathematical Engineering, Stanford University, Stanford, United States; [6]Division of CryoEM and Bioimaging, SSRL, SLAC National Accelerator Laboratory, Stanford University, Stanford, United States; [7]Department of Bioengineering and Department of Microbiology and Immunology, Stanford University, Stanford, United States

*For correspondence:
maduke@stanford.edu

**Abstract** CLC-2 is a voltage-gated chloride channel that contributes to electrical excitability and ion homeostasis in many different tissues. Among the nine mammalian CLC homologs, CLC-2 is uniquely activated by hyperpolarization, rather than depolarization, of the plasma membrane. The molecular basis for the divergence in polarity of voltage gating among closely related homologs has been a long-standing mystery, in part because few CLC channel structures are available. Here, we report cryoEM structures of human CLC-2 at 2.46 – 2.76 Å, in the presence and absence of the selective inhibitor AK-42. AK-42 binds within the extracellular entryway of the Cl⁻-permeation pathway, occupying a pocket previously proposed through computational docking studies. In the apo structure, we observed two distinct conformations involving rotation of one of the cytoplasmic C-terminal domains (CTDs). In the absence of CTD rotation, an intracellular N-terminal 15-residue hairpin peptide nestles against the TM domain to physically occlude the Cl⁻-permeation pathway. This peptide is highly conserved among species variants of CLC-2 but is not present in other CLC homologs. Previous studies suggested that the N-terminal domain of CLC-2 influences channel properties via a "ball-and-chain" gating mechanism, but conflicting data cast doubt on such a mechanism, and thus the structure of the N-terminal domain and its interaction with the channel has been uncertain. Through electrophysiological studies of an N-terminal deletion mutant lacking the 15-residue hairpin peptide, we support a model in which the N-terminal hairpin of CLC-2 stabilizes a closed state of the channel by blocking the cytoplasmic Cl⁻-permeation pathway.

## eLife assessment

This study examines the human voltage-gated chloride channel CLC-2. A combination of cryo-EM, electrophysiology, and computational analysis provides **compelling** support for a "ball and chain" mechanism of inactivation. This and other findings regarding the gating and inhibition mechanisms of the channel are of **fundamental** interest to ion channel physiologists.

## Introduction

Ion transport across the cell membrane establishes and exploits ion gradients to store energy and regulate electrical signaling. Ion transport is catalyzed by two classes of membrane proteins: ion channels and active transporters. The CLC 'Chloride Channel' family encompasses both major ion-transport mechanisms: half of the CLC homologs are electrodiffusive chloride ($Cl^-$) channels, and half are secondary active transporters that stoichiometrically exchange $Cl^-$ for protons ($H^+$) (*Jentsch and Pusch, 2018*). The occurrence of two mechanisms in one family suggests they operate by variations on a common theme (*Miller, 2006*), and experimental results revealed that CLC channels are 'broken' transporters (*Lísal and Maduke, 2008*).

In both CLC channels and transporters, a conserved glutamate residue referred to as the 'gating glutamate' ($E_{gate}$) plays a central mechanistic role. High-resolution CLC transporter structures show that this residue can adopt four different conformations (*Dutzler et al., 2002*; *Dutzler et al., 2003*; *Feng et al., 2010*; *Chavan et al., 2020*; *Schrecker et al., 2020*, *Figure 1A*), which contribute to the $Cl^-/H^+$ exchange mechanism (*Accardi and Miller, 2004*; *Feng et al., 2012*; *Chavan et al., 2020*; *Leisle et al., 2020*). Due to the small number of CLC channel structures relative to transporters, characterization of the central role for the $E_{gate}$ residue in CLC channels has primarily relied on extensive functional studies of channel gating rather than observed $E_{gate}$ side-chain conformations (*Jentsch and Pusch, 2018*). The first CLC channel structure reported was of bovine CLC-K (*Park et al., 2017*), a kidney-expressed homolog that is the lone CLC homolog lacking the $E_{gate}$ (*Kieferle et al., 1994*). The only other CLC channel structures reported prior to 2023 are of the human CLC-1: one of these

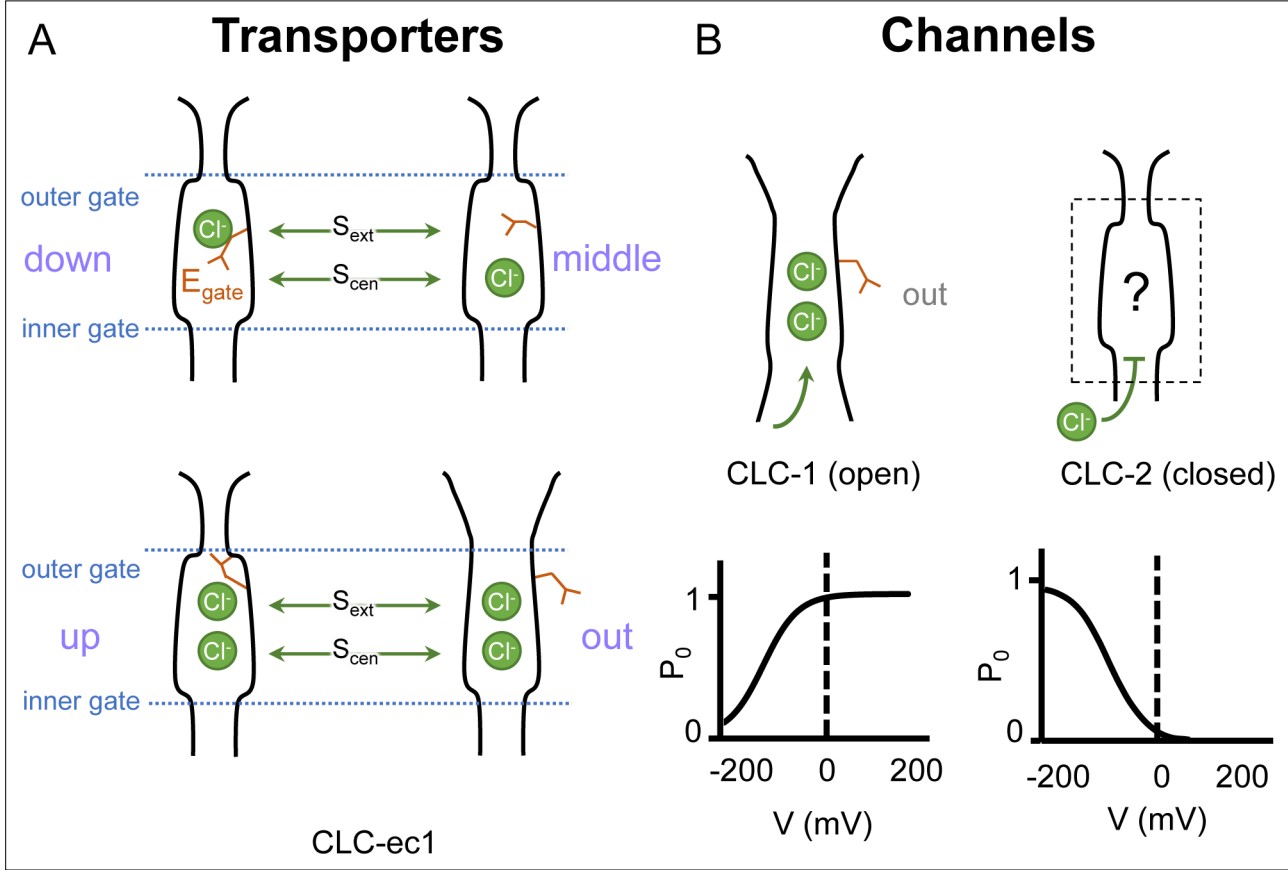

**Figure 1.** Gating conformations in CLC transporters and channels. (**A**) Cartoon depictions of the CLC anion permeation pathway. The 'gating glutamate' ($E_{gate}$) can occupy anion-binding sites within the permeation pathway ('down' and 'middle' conformations, occupying the central ($S_{cen}$) and external ($S_{ext}$) anion-binding sites, respectively) or away from these sites ('up' and 'out' conformations). Coupling of $E_{gate}$ movement to $Cl^-/H^+$ binding/unbinding and global conformational change generates 2:1 $Cl^-/H^+$ exchange. (**B**) $E_{gate}$ is required for voltage-dependent gating in CLC channels but cryoEM density for this residue has been observed only in CLC-1 (PDB ID: 6coy), in the 'out' position. In contrast to CLC-1, which is predominantly open at zero mV, CLC-2 is predominantly closed. The open probability ($P_o$) versus voltage ($V$) traces were drawn based on published data reported for CLC-1 (*Lísal and Maduke, 2008*) and CLC-2 (*Stölting et al., 2013*).

cryoEM structures shows $E_{gate}$ in the 'out' position (*Park and MacKinnon, 2018*) while the other lacked density for $E_{gate}$ (*Wang et al., 2019*). The $E_{gate}$ 'out' position was originally posited to be exclusive to the CLC channel homologs, as its positioning off to the side widens the $Cl^-$ permeation pathway and, in CLC-1, results in a continuous pathway (*Park and MacKinnon, 2018*). Such a continuous pathway precludes exchange coupling (*Jardetzky, 1966*; *Miller, 2021*). The open $Cl^-$ permeation pathway in the CLC-1 structure is consistent with the known functional properties of CLC-1 in the absence of an applied transmembrane (TM) voltage. CLC-1 is a voltage-gated channel that is predominantly in the open state at zero mV (*Altamura et al., 2020*, *Figure 1B*). At negative TM voltages that favor CLC-1 closing, the $E_{gate}$ side chain presumably moves to either the 'middle' or 'down' positions, or both, to occlude the channel pore.

CLC-2, like CLC-1, is a voltage-gated channel, but with opposite voltage dependence, opening instead of closing in response to hyperpolarization (*Thiemann et al., 1992*; *Wang et al., 2017*). CLC-2 chloride channels are expressed in every mammalian organ, where they play diverse physiological functions (*Thiemann et al., 1992*; *Lein et al., 2007*; *Duan, 2013*; *Stölting et al., 2014*; *Jentsch and Pusch, 2018*). Perturbation of CLC-2 function is associated with human disease including leukodystrophies (*Jentsch and Pusch, 2018*), blindness (*Xu et al., 2023*), primary aldosteronism (*Fernandes-Rosa et al., 2018*; *Scholl et al., 2018*), and atrial fibrillation (*Hansen et al., 2020*). Despite their importance in human physiology, our understanding of CLC-2 molecular and physiological mechanisms is incomplete. A mystery concerning CLC-2 is why electrophysiological recordings of natively expressed channels reveal tremendous variation in voltage-dependent gating. CLC-2 currents are typically inwardly rectifying and activated by hyperpolarization, but the kinetics and degree of rectification are highly variable (*Bösl et al., 2001*; *Jentsch and Pusch, 2018*). In some cases, this variability is understood – for example interactions between CLC-2 and the astrocytic GlialCAM protein yields voltage-independent currents (*Jeworutzki et al., 2012*; *Maduke and Reimer, 2012*). Yet many questions about CLC-2 gating remain.

Like CLC-1, CLC-2 gating depends critically on $E_{gate}$ (*Niemeyer et al., 2003*; *Cederholm et al., 2010*). At zero mV, CLC-2 is predominantly in the closed state (*Stölting et al., 2013*; *Stölting et al., 2014*; *Jentsch and Pusch, 2018*), presumably with $E_{gate}$ blocking the pore. Such pore localization for $E_{gate}$ was modeled in a recent CLC-2 cryoEM structure (*Ma et al., 2023*) however, experimental density for the side chain was lacking, and therefore the precise structure of this gate remains uncertain (*Figure 1B*). Structures with improved resolution will be crucial to understanding CLC-2 gating, which involves interactions between $E_{gate}$, $Cl^-$, and $H^+$, with differential effects of external vs internal ions (*Pusch et al., 1999*; *Niemeyer et al., 2003*; *Sánchez-Rodríguez et al., 2010*; *Sánchez-Rodríguez et al., 2012*; *De Jesús-Pérez et al., 2016*). In addition to the $E_{gate}$ mechanism, a 'ball-and-chain' gating mechanism, in which the tethered N-terminus blocks the pore, has been proposed for CLC-2 (*Gründer et al., 1992*). However, this idea has been controversial (*Varela et al., 2002*) and the mechanism by which mutations in the N-terminal domain influence gating remains unknown. Because a structural foundation for understanding CLC-2 gating has been lacking, we determined cryoEM structures of CLC-2, in the presence and absence of the selective inhibitor AK-42 (*Koster et al., 2020*). The structures and complementary electrophysiology data presented here provide atomic insight into an N-terminal hairpin gating mechanism for CLC-2 channels.

## Results

### Overview of cryoEM structure determination

Human CLC-2 (hCLC-2) protein was purified from cultured HEK293 GnTI⁻ cells and further examined by cryoEM single-particle analysis. Selected two-dimensional class averages showed a clear TM domain embedded in a detergent micelle and features indicating an intracellular domain. Further refinement revealed a well-resolved TM domain density at 2.46 Å, 'CLC2-TM' (*Figure 2*; *Figure 2—figure supplements 1–3*). Density for the C-terminal intracellular domain (CTD) is less well resolved. Such relatively weak density for the CTD relative to the TM domain was also observed in the cryoEM analysis of human CLC-1 (*Park and MacKinnon, 2018*; *Wang et al., 2019*), which is hCLC-2's closest homolog. The CTD of CLC-0, a CLC-1 ortholog from *Torpedo mamorata* (*Steinmeyer et al., 1991*) displays large movements in spectroscopic studies (*Bykova et al., 2006*; *Alioth et al., 2007*). Thus, conformational flexibility in the CTD appears to be a shared feature within this branch of the CLC

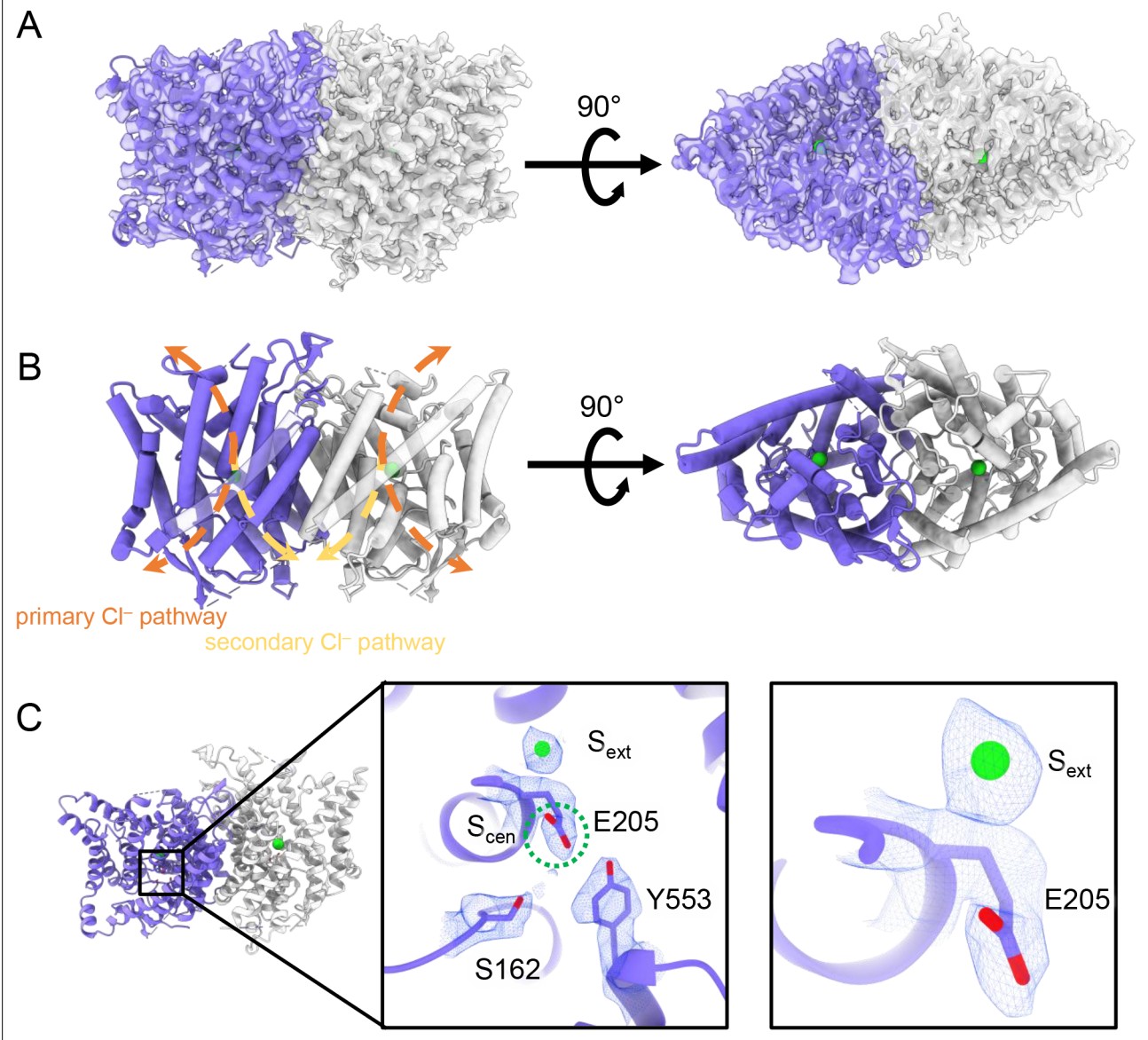

**Figure 2.** CryoEM structure of the human CLC-2 channel. Overall structure of the transmembrane domain (CLC2-TM) at 2.46 Å. The identical subunits of the homodimer are shown in purple and gray; $Cl^-$ ions are shown as green balls. (**A**) CryoEM density map with model overlay (contour level: 0.71, 5.5 σ) and (**B**) model only with dashed arrows indicating the locations of pathways detected by Caver: orange, primary $Cl^-$ pathway common to all CLCs; yellow, secondary $Cl^-$ pathway detected in CLC-2 and CLC-1. Details of the pathways are presented in *Figure 3*. The secondary pathway, whose function is not yet known, was so named when first observed in the CLC-1 structure (*Park and MacKinnon, 2018*). (**C**) Zoomed-in views showing cryoEM density overlay of bound $Cl^-$ and key residues – $E_{gate}$ (E205) and inner-gate residues $Ser_C$ (S162) and $Tyr_C$ (Y553). $S_{cen}$ is indicated with a green dashed circle. Contour level: 1.1, 7.2 σ (middle panel); 0.97, 6.3 σ (right panel).

The online version of this article includes the following figure supplement(s) for figure 2:

**Figure supplement 1.** Micrograph, 2D classes, and structure validation of CLC2-TM.

**Figure supplement 2.** CryoEM workflow of the CLC2 single-particle cryoEM data processing.

**Figure supplement 3.** Helix map of CLC2-TM.

**Figure supplement 4.** $E_{gate}$ position and $Cl^-$-binding sites ($S_{ext}$ and $S_{cen}$).

family. To further characterize CTD map density, 3D classification was performed. Among 10 3D classes identified, two showed significantly improved density for the CTD. With non-uniform refinement and further local refinement, we determined two conformations of CLC-2 structures at 2.75 and 2.76 Å, respectively (*Figure 2—figure supplement 2*).

## Structure of the TM domain

The TM region of CLC-2 displays a typical CLC family symmetric homodimeric structure, with each subunit containing an independent $Cl^-$ pathway (*Figure 2A, B*). In the cryoEM map, clear density for the $E_{gate}$ residue (E205 in CLC-2) reveals it occupying the 'down' position, at the $S_{cen}$ $Cl^-$-binding site (*Figure 2C*). This downward $E_{gate}$ positioning has previously been observed only in the X-ray crystallographic structure of the *C. merolae* CLC transporter homolog (cmCLC) (*Feng et al., 2010*), but not in any CLC channel homolog (*Figure 2—figure supplement 4A, B*). In the recent CLC-2 structure by Ma et al., $E_{gate}$ lacked density and was modeled in two different conformations (*Ma et al., 2023*, *Figure 2—figure supplement 4C*). Our cryoEM map also reveals a density feature at the $S_{ext}$ site, consistent with a bound $Cl^-$ ion (*Figure 2C*). The ion at the $S_{ext}$ site is stabilized by surrounding backbone nitrogen atoms, a characteristic feature of CLCs (*Dutzler et al., 2003*; *Picollo et al., 2009*; *Feng et al., 2010*; *Park and MacKinnon, 2018*).

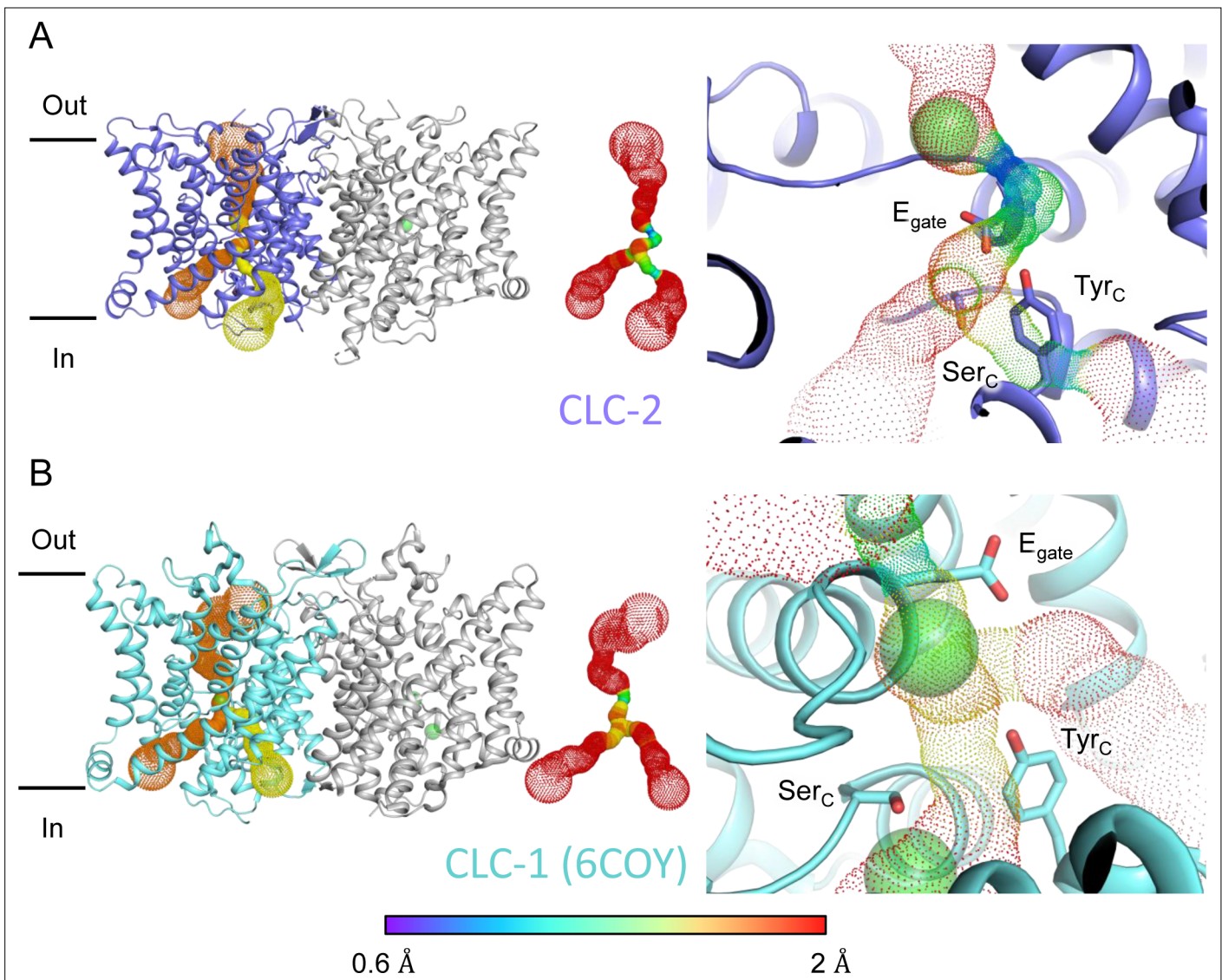

**Figure 3.** Pore profile detected in CLC-2. (**A**) Left: The primary (orange) and secondary (yellow) pore detected in CLC-2 using Caver. Middle: The detected pore radii are displayed in dots as a color map. Right: Zoomed-in view of the restricted (narrowest) site of the pore profile, using the same coloring as in the middle panel. $E_{gate}$, $Ser_C$, and $Tyr_C$ are shown as sticks. (**B**) Pore profile for CLC-1 (PDB ID: 6coy), with display as in panel A.

The online version of this article includes the following figure supplement(s) for figure 3:

**Figure supplement 1.** $Cl^-$ pathway in CLC-2 and CLC-1.

**Figure supplement 2.** Comparison of primary and secondary $Cl^-$ pathways with transporter $Cl^-$ and $H^+$ pathways.

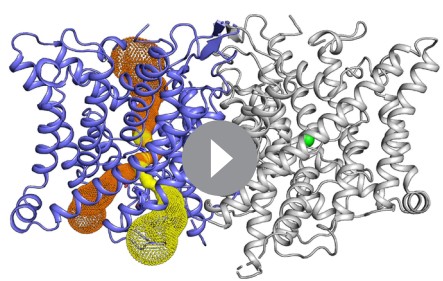

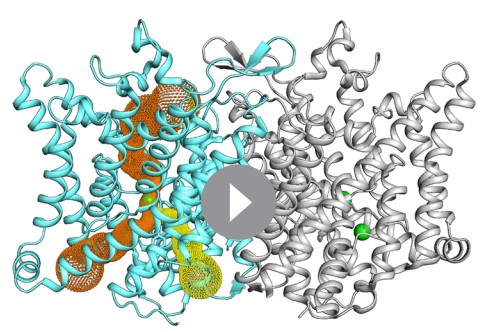

**Video 1.** CLC-2 pore detected using Caver.
https://elifesciences.org/articles/90648/figures#video1

**Video 2.** CLC-1 pore detected using Caver.
https://elifesciences.org/articles/90648/figures#video2

Two lines of evidence point to the structure representing a closed state, as expected at zero mV. First, $E_{gate}$ physically occupies the $S_{cen}$ Cl⁻-binding site (*Figures 1A and 2C*), thus blocking the permeation pathway. Second, we used Caver (*Chovancova et al., 2012*) to detect pathways through the protein. A continuous pathway through the protein could only be detected with a very small probe radius of 0.5 Å. In this case, the route detected follows the canonical Cl⁻ transport pathway from the extracellular side to $S_{cen}$, but then circumnavigates $E_{gate}$ to reach the intracellular pathway (*Figure 3A*, *Video 1*). The pinch point of ~0.7 Å near the $E_{gate}$ residue is too narrow for a Cl⁻ ion and thus indicates a closed conformational state. The CLC-2 pathway is constricted compared to that in CLC-1, where the location of $E_{gate}$ in the 'out' position permits a wider Cl⁻ diffusion pathway (*Figure 3B*, *Video 2*). Close-up views of the canonical Cl⁻ pathway in CLC-2 and CLC-1 are shown in *Figure 3—figure supplement 1*. The two lines of evidence just discussed – $E_{gate}$ at the $S_{cen}$ site and the Caver pinch point – argue that $E_{gate}$ is in the deprotonated state: (1) the location of $E_{gate}$ in an anion-binding site is most consistent with the side chain being negatively charged; (2) neutralization of $E_{gate}$ leads to an open-channel phenotype (*Niemeyer et al., 2003*; *de Santiago et al., 2005*), which is incompatible with the observed pinched-closed pore.

At the CLC inner-gate region, near conserved residues $Tyr_C$ (Y553) and $Ser_C$ (S162), the canonical (primary) Cl⁻ permeation pathway is similar for CLC-2 and CLC-1 and wider than that seen in the CLC transporters (*Figure 3—figure supplement 2A, B*, *Video 3*). CLC-2 also displays a bifurcated pore, with a secondary pathway to the intracellular side, distinct from the primary pathway common to CLC family members. The secondary pathway is shared with CLC-1 but not seen in other CLC homologs of known structure. While the functional role of this distinctive secondary pore is not yet known, it is intriguing to note it is located near the region thought to act as a conduit for H⁺ in the CLC transporters (*Figure 3—figure supplement 2C*).

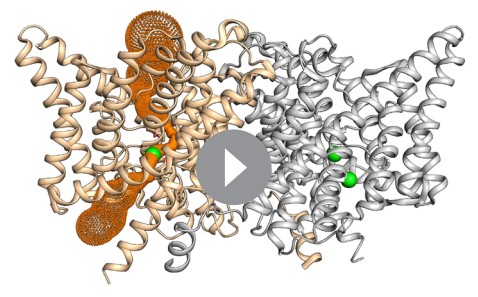

**Video 3.** CLC-ec1 caver-detected pathway.
https://elifesciences.org/articles/90648/figures#video3

## Two conformations of the intracellular domain

All eukaryotic CLCs contain a pair of 'cystathionine β-synthase' (CBS) domains in their intracellular C-terminal domain (CTD) (*Jentsch and Pusch, 2018*). CBS domains are found in a wide variety of functionally unrelated protein families, in all kingdoms of life, where they serve to regulate the activity of the attached protein (*Ereño-Orbea et al., 2013*; *Anashkin et al., 2017*). In the CLCs, mutations in or near the CBS domains alter channel gating and are associated with human disease, including myotonia, Dent's disease, Bartter syndrome, osteopetrosis, leukoencephalopathy, retinal degeneration, lysosomal storage

disease, and neurodevelopmental disorders (*Estévez et al., 2004*; *Giorgio et al., 2017*; *Jentsch and Pusch, 2018*; *Guo et al., 2019*; *Nicoli et al., 2019*; *Altamura et al., 2020*; *Gianesello et al., 2020*; *Duncan et al., 2021*; *Zifarelli, 2022*; *Palmer et al., 2023*; *Xu et al., 2023*). Structurally, the CLC CBS domains interact across subunits, forming dimers of dimers and thus increasing the overall interaction area of the homodimer (*Feng et al., 2010*; *Park et al., 2017*; *Park and MacKinnon, 2018*; *Wang et al., 2019*; *Schrecker et al., 2020*). Structural rearrangement of the CTDs is thought to occur during channel gating (*Yamada et al., 2016*; *Jentsch and Pusch, 2018*) and has been observed spectroscopically (*Bykova et al., 2006*; *Ma et al., 2011*), but no rearrangements have yet been observed at high resolution. The physiological relevance of CTD rearrangements is suggested by disease-causing mutations that alter channel gating (*Estévez et al., 2004*; *Brenes et al., 2023*).

During the 3D classification data processing with a C1 reconstruction (no symmetry imposed), two distinct conformations were separated from the same CLC-2 dataset (*Figure 2—figure supplement 2*). Structural comparison of these two classes reveals that they are very similar in the TM domain, with root-mean-square deviation (RMSD) of 0.58 and 0.42 Å for the two subunits. The main structural differences occur at the intracellular domain, with one class exhibiting a symmetric arrangement and the other, asymmetric. These two conformations are named as CLC2-CTDsym and CLC2-CTDasym, respectively (*Figure 4*, *Figure 4—figure supplements 1 and 2*). In both classes, the CBS domains can be fit to the density; however, residues 645–743 in the CLC-2 CTD are not resolved and thus are omitted from the models and from the *Q*-score calculation (*Pintilie et al., 2020*, *Figure 4—figure supplement 1*). In the asymmetric arrangement (CLC2-CTDasym), one of the CTDs rotates toward the TM region approximately 35° relative to the conformation in the symmetric structure (*Figure 4B*). Structural comparison focused on individual CTDs within the two classes reveals low RMSD values (~1 Å), which indicates that the CTD undergoes a rigid body movement to accomplish the conformational change (*Figure 4C*, *Video 4*). Molecular dynamics (MD) simulations confirm conformational dynamics consistent with a domain rotation (*Figure 4—figure supplement 3*). In both structures, the orientation of the CTD relative to the membrane is substantially different from that observed in other CLC structures (*Figure 4—figure supplement 4*).

## The hairpin structure corresponds to the N-terminal region

Another new feature of the determined CLC-2 structure is a hairpin density blocking the main Cl⁻ pathway on the intracellular side (*Figure 4A*). This feature is not continuous with any other protein density in the map. In theory, such density could arise from a protein sequence on CLC-2 (from the unstructured N- or C-terminal regions) or from a separate protein carried along in the purification. Given the history of studies on CLC-2 gating, we first considered the CLC-2 N-terminal region.

Shortly after the initial cloning of CLC-2 (*Thiemann et al., 1992*), experiments on CLC-2 expressed in *Xenopus* oocytes demonstrated that deletion of an 18-residue segment of the N-terminus yielded a constitutively open channel, lacking the characteristic voltage-dependent gating of CLC-2, and that transplantation of this segment to the CTD restored gating (*Gründer et al., 1992*). Furthermore, mutation of certain residues in the TM domain (at a putative 'receptor' site for the N-terminal blockade) also yielded an open-channel phenotype (*Jordt and Jentsch, 1997*). Based on these results, it was argued that CLC-2 has a 'ball-and-chain' type gating mechanism reminiscent of that occurring in the *Shaker* potassium channel (*Zagotta et al., 1990*). However, experiments performed using the inside-out patch-clamp configuration (*Pusch et al., 1999*) or using whole-cell recording with CLC-2 expressed in mammalian cells (*Varela et al., 2002*) failed to recapitulate the gating phenotype and called into question a ball-and-chain mechanism. Thus, while mutations in the N-terminus of CLC-2 influence gating, the precise mechanism-of-action has been uncertain.

Our cryoEM map allowed continuous modeling of the CLC-2 sequence starting from the first TM helix (residue 88). The lack of sufficient density for modeling residues 1–87 suggests that the N terminal cytoplasmic region of CLC-2 is intrinsically disordered. However, our observation of a resolvable hairpin density at the intracellular pore, together with the previous hypothesis of N-terminal ball-and-chain gating for CLC-2, prompted us to consider whether the hairpin density corresponds to N-terminal residues. We tested several candidate fragments from the CLC-2 N-terminal region and found that the hairpin density fits perfectly with residues corresponding to 14–28 (*Figure 5A*). Inspection of the *Q*-score validation plot for these residues shows they have higher values than the *Q*-score expected for the corresponding resolution (*Figure 5B*), which indicates reliable model building at

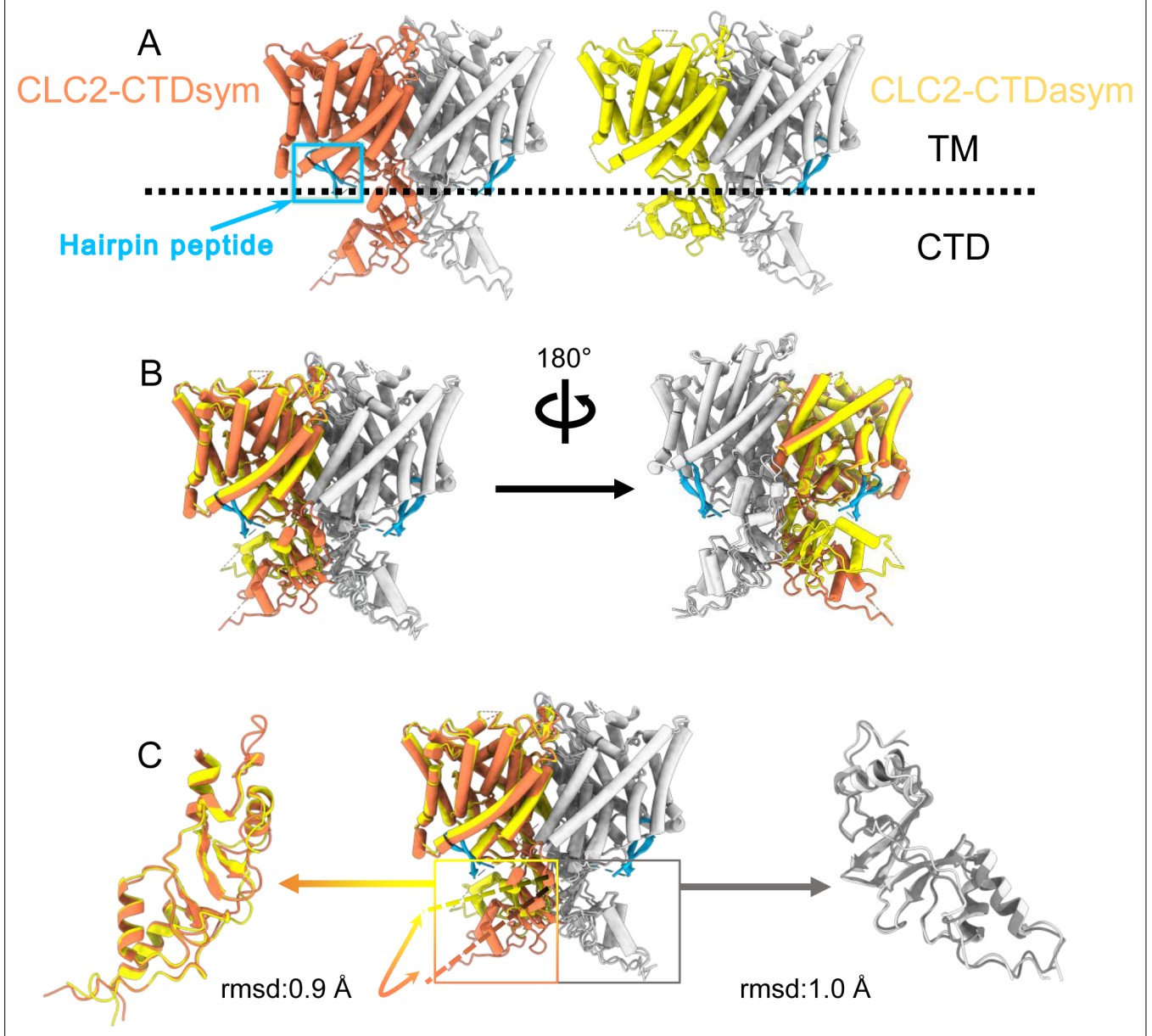

**Figure 4.** Different conformations of the C-terminal domain (CTD). (**A**) Overall structure of the two CLC-2 conformations. Left: CLC2-CTDsym at 2.75 Å. Right: CLC2-CTDasym at 2.76 Å. The subunits of the homodimer are shown in orange (CLC2-CTDsym) or yellow (CLC2-CTDasym) and gray (for both). The subunits adopting different CTD conformations are shown in orange and yellow, while the subunits sharing the same CTD conformation are shown in gray. The dashed line demarcates the transmembrane (TM) and cytoplasmic (CTD) domains. A hairpin structure detected at the intracellular pore is shown in blue. (**B**) Overlay of two conformations. (**C**) Hinge movement of the CTD between two conformations. The middle panel shows an overlay of CLC2-CTDsym (orange and gray) and CLC1-CTDasym (yellow and gray), with the CTDs highlighted in boxes. For the subunits that adopt different conformations (orange and yellow), the double-headed arrow indicates the change between the two conformations. The root-mean-square deviation (RMSD) between the subunits shown in orange/yellow is 6.9 Å, while the RMSD for the subunits shown in gray is only 1.3 Å. Alignments of only the CTDs are shown at left and right; in these alignments, the low RSMD values (0.9 and 1.0 Å, respectively) indicate that the CTDs undergo a hinge movement.

The online version of this article includes the following figure supplement(s) for figure 4:

**Figure supplement 1.** Structure validation of CLC2-CTDsym and CLC2-CTDasym.

**Figure supplement 2.** CryoEM density maps for the two C-terminal domain (CTD)-containing CLC-2 conformations, overlaid with structural models.

**Figure supplement 3.** Molecular dynamics (MD) analysis indicates conformational flexibility of the C-terminal domain (CTD).

**Figure supplement 4.** Comparison of C-terminal domain (CTD) orientation with respect to the transmembrane (TM) domain.

**Figure supplement 5.** ATP-binding site comparison between CLC-2 and CLC-5.

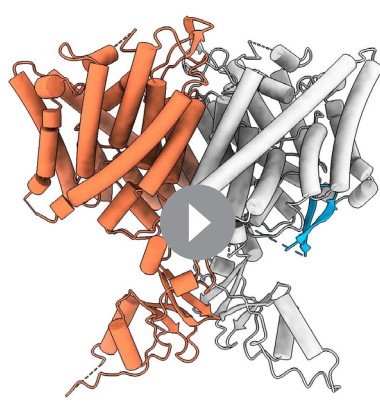

**Video 4.** Rigid body movement from CLC2-CTDsym to CLC2-CTDasym.
https://elifesciences.org/articles/90648/figures#video4

this resolution (*Pintilie et al., 2020*). In contrast, the best-fitting peptide from disordered regions of the C-terminal cytoplasmic region yielded poorer *Q*-scores, generally lower than the *Q*-score expected for the corresponding resolution (*Figure 5—figure supplement 1*). The N-terminal residues corresponding to the hairpin peptide are conserved throughout the mammalian CLC-2 homologs and partially conserved in *Drosophila* CLC-2 (*Figure 5—figure supplement 2*).

Interestingly, the presence of the N-terminal hairpin is associated with conformational changes in the CTD and TM domains. In CLC2-CTDsym, N-terminal density was observed on both subunits. In CLC2-CTDasym, only the subunit whose CTD adopts the 'symmetric' orientation (seen in CLC2-CTDsym) contains the hairpin density; in the subunit with the rotated CTD, no hairpin density is observed (*Figure 4*, *Figure 5A*). Along with the absence of hairpin density, the subunit with the rotated CTD exhibits a difference in the linker connecting TM helices J and K, which is poorly resolved and insufficient for model building. In contrast, the subunits with the hairpin density contain reliable density in the JK-linker region, allowing confident modeling of an alpha helix ('JK-helix', *Figure 5A*). This observation suggests that the JK-helix stabilizes the N-terminal hairpin peptide. Indeed, residues on the JK-helix interact directly with residues at the peptide's hairpin turn (*Figure 5C*). Interactions with other regions in the TM domain (near helices C, D, J, and R) appear to further stabilize the hairpin in its binding site. In addition, the negative surface potential of the hairpin likely contributes to its interaction with the positive TM domain (*Figure 5—figure supplement 3*; *Video 5*). MD simulations help separately validate the assignment and stability of the hairpin: in MD simulations initiated from the structure, we found that the N-terminus remained tightly associated with the channel in all simulations (*Figure 5—figure supplement 4*).

## Functional support for pore block by the N-terminal hairpin structure

The hairpin structure appears to directly block the pore (*Figure 5—figure supplement 5*). To evaluate the potential functional effect of pore block by the N-terminal hairpin, we performed whole-cell patch-clamp recording on CLC-2, both WT and a deletion mutant lacking the hairpin residues (Delta-N), expressed in CHO cells. Delta-N CLC-2 exhibits currents that display less rectification and faster activation kinetics than WT CLC-2 (*Figure 6A, B*, *Figure 6—figure supplement 1*). This gating phenotype is intermediate between the non-rectifying phenotype of N-deletion constructs from whole-cell recordings in oocytes (*Jordt and Jentsch, 1997*) and the highly rectifying phenotype from whole-cell recordings in HEK 293 cells (*Varela et al., 2002*). The Discussion section addresses the variability observed in CLC-2 current recordings. As was reported for N-deletion currents in oocytes, our Delta-N currents are substantially larger than WT currents, ~sixfold on average when identical transfection conditions are used (*Figure 6C*). This difference cannot be explained by a difference in plasma-membrane expression, as surface biotinylation experiments revealed that expression levels of the two constructs are within 30% of one another (*Figure 6D, E*).

Strikingly, WT but not Delta-N CLC-2 currents 'run-up' during recording sessions. In patch-clamp recording experiments, the term 'run-up' describes a progressive increase in current that occurs after the initial measurement. To quantify CLC-2 run-up, we developed the following protocol: (1) on acquisition of whole-cell recording mode, an initial voltage-family of currents was recorded, as in *Figure 6A*; (2) from a holding potential of 0 mV, a single 1-s test pulse to −100 mV was repeated every 5 s to monitor time-dependent changes in current amplitude over 5 min, after which a second voltage-family of currents was recorded; (3) 300 nM AK-42 was added to selectively block all CLC-2 current (*Koster et al., 2020*), and the steady-state inhibition current was used for leak subtraction. In contrast

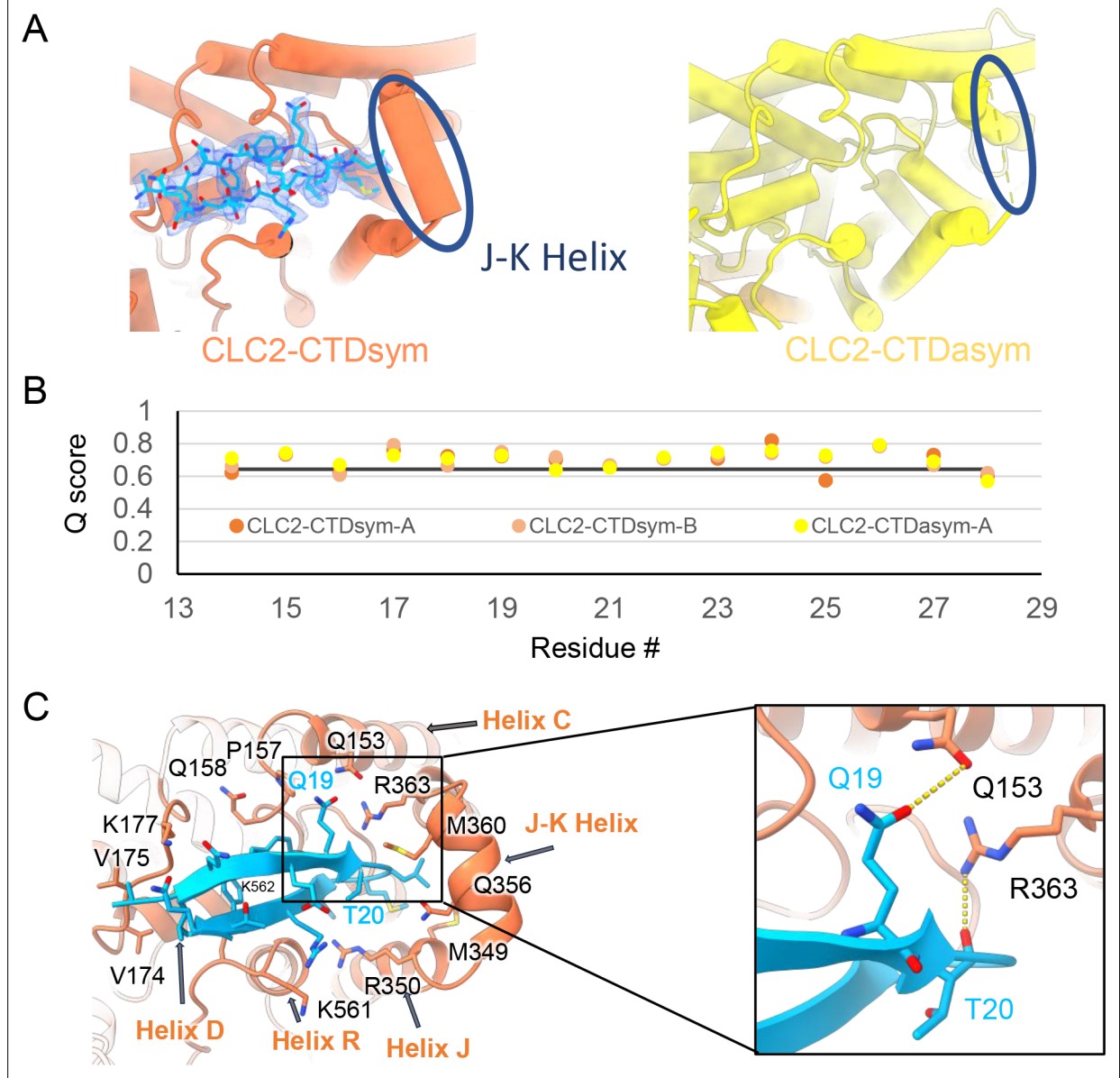

**Figure 5.** The blocking hairpin structure corresponds to an N-terminal sequence of CLC-2. (**A**) Left: The hairpin structure seen in the cryoEM density (blue mesh) fits N-terminal residues 14–28 (blue). Right: The subunit in CLC2-CTDasym structure that contains a C-terminal domain (CTD) with rotated orientation lacks density for the hairpin structure. In this structure, the residues linking transmembrane (TM) helices J and K are not resolved. The dark blue circles indicate this unresolved region (right panel) and the corresponding resolved J-K helix that occurs in the other subunits. (Contour level: 0.76, 5.5 σ.) (**B**) Model validation using Q-scores for the N-terminal hairpin: orange and light orange for subunits A and B of CLC2-CTDsym subunit; yellow for subunit A (the hairpin-containing subunit) of CLC2-CTDasym. The black line represents the expected Q-score at 2.75 Å based on the correlation between Q-scores and map resolution. (**C**) Residues on the TM domain interact with the N-terminal hairpin. The J-K helical linker and TM helices C, D, R, and J are labeled with arrows. Inset: Zoomed-in view of two hydrogen bonds formed between the hairpin structure and the TM domain: Q19–Q153 and T20–R363. Mutation of these two residues (Q25 and T26 in rat CLC-2) generated an open-channel phenotype for CLC-2 expressed in *Xenopus* oocytes (*Gründer et al., 1992*).

The online version of this article includes the following figure supplement(s) for figure 5:

**Figure supplement 1.** Comparison of peptide candidates fitting into the hairpin density.

**Figure supplement 2.** N-terminal hairpin sequence is conserved in CLC-2, but not other CLCs.

**Figure supplement 3.** Electrostatic map of the N-terminal hairpin-blocking site.

**Figure supplement 4.** Molecular dynamics (MD) simulations confirm the N-terminal hairpin structure is stable.

**Figure supplement 5.** The N-terminal hairpin blocks the primary Cl⁻ pore in CLC-2.

**Figure supplement 6.** Mapping mutant data onto the cryoEM structure.

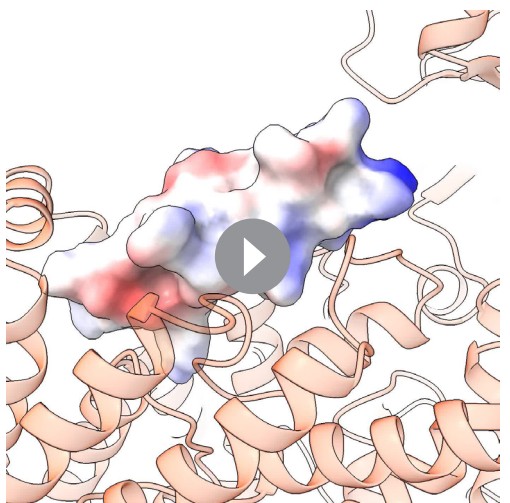

**Video 5.** Hairpin surface potential.
https://elifesciences.org/articles/90648/figures#video5

to chloride-channel inhibitors historically used in electrophysiological experiments (*Sepela and Sack, 2018*), AK-42 is potent and highly selective for CLC-2 over other anion channels (*Koster et al., 2020*). Representative current traces and summary data for this set of experiments are shown in *Figure 6F, G*; individual *I–V* traces are shown in *Figure 6—figure supplement 1*. The increase in WT but not Delta-N CLC-2 currents over time is consistent with an 'unblocking' of the pore by release of the N-terminal hairpin peptide.

The increase in WT currents is accompanied by an apparent decrease in channel rectification, toward the level displayed by Delta-N CLC-2. WT CLC-2 currents recorded in heterologous and native expression systems are generally highly inwardly rectifying. Analogous to a rectifier in an electrical circuit, WT CLC-2 acts as a directional current valve, allowing much larger absolute currents to flow at negative membrane potentials compared to positive ones. At the beginning of each recording, our WT CLC-2 currents display such inward rectification, but this characteristic appears to diminish during 5 min of repeated voltage pulsing to −100 mV, as substantial currents at positive voltages appear. For the experiment shown in *Figure 6F*, the steady-state current at +80 mV increases from 161 pA to 902 pA, almost sixfold, compared to a 2.4-fold increase at −100 mV. This observation is surprising because WT CLC-2 has been characterized as passing little to no current at positive voltages (*Park et al., 1998*; *Arreola et al., 2002*). We initially considered that the increase in current at +80 mV could be due to the patch-clamp seal becoming leaky over time or that a background channel is being activated; however, such 'leak' currents would not be expected to be blocked by the small-molecule inhibitor AK-42, which is highly selective for CLC-2 over all other CLCs and anion channels (*Koster et al., 2020*). AK-42 block of the augmented +80 mV currents seen at the end of each WT CLC-2 recording supports the idea that the hairpin peptide may contribute to the inward rectification of CLC-2 (*Figure 6F*, *Figure 6—figure supplement 2*).

## AK-42-bound structure

AK-42's selectivity empowered our biophysical studies of CLC-2 and will also be invaluable for studies of CLC-2's physiological function. We determined the structure of CLC-2 in complex with AK-42 (*Figure 7*). While the TM domain could be resolved to 2.7 Å, the CTD exhibited poor density compared to the apo sample and did not allow us to confidently model the CTD when masking the whole protein for refinement. To improve the density quality, we masked the CTD only for refinement; this masking provided an acceptable density map at 4 Å and allowed us to confidently dock the CTD from apo structure (*Figure 7—figure supplements 1–4*).

Like the apo CLC-2 structure, the TM domain of CLC2-TM-AK42 is well resolved with clear densities for AK-42 and for CLC-2 side chains, including the $E_{gate}$ side chain (*Figure 7—figure supplements 5 and 6*). The overall density for CLC2-TM-AK42 is very similar to that observed in apo CLC-2, but with an extra density that fits very well to the AK-42 molecule (*Figure 7B*, *Video 6*). Residues in the AK-42-binding pocket exhibit little conformational change between the apo and AK-42-bound CLC-2 structures. The exception is residue M460, which moves ~3 Å to avoid a clash with AK-42's phenyl ring (*Figure 7—figure supplement 7*). Located at the extracellular side of Cl⁻ pathway, the bound AK-42 appears to directly block Cl⁻ permeation (*Figure 7C*). This AK-42-binding pocket is the same as observed in the docking prediction we previously made using a CLC-2 homology model (*Koster et al., 2020*). In that docking study, the top docking poses varied in detail, and therefore the specific interactions were not certain, but the general position of the binding site is confirmed by our cryoEM structure. Two conformations were found to fit the density well with the carboxyl group on opposite sides. They were tested using the Quantum Mechanical Restraints (QMR) method (*Liebschner et al.,*

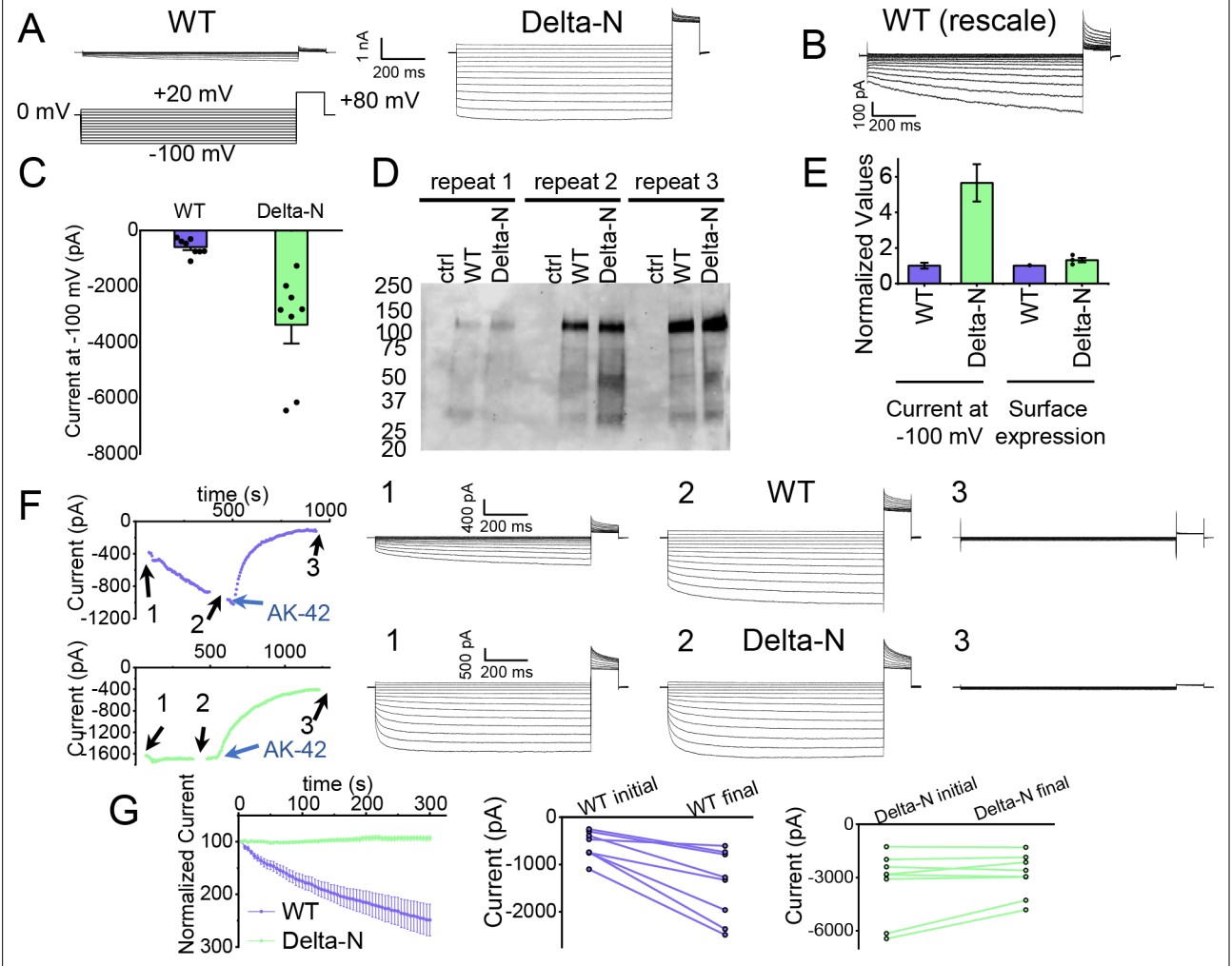

**Figure 6.** Patch-clamp experiments support CLC-2 channel block by the N-terminal hairpin structure. (**A**) Representative currents from WT and Delta-N CLC-2, recorded using the whole-cell patch-clamp configuration in response to the voltage protocol shown. (**B**) WT CLC-2 recording from panel A, shown on an expanded scale. (**C**) Summary of current levels measured for WT and Delta-N CLC-2 at the end of the 1 s voltage pulse to −100 mV. Data are from six independent transfection samples, in each case with WT and Delta-N recorded on the same day following transfection. WT: −600 ± 102 pA (standard error of the mean [SEM], $n = 8$); Delta-N: −3300 ± 630 pA (SEM, $n = 8$). (**D**) Western blot detection of biotinylated surface-expressed CLC-2 from three independent experiments. (**E**) Summary data for electrophysiology and surface biotinylation experiments. Points representing individual experiments for Delta-N surface biotinylation (each normalized to WT) are shown. Individual data points for the electrophysiology experiments are shown in panel C. (**F**) Representative examples of experiments to evaluate current run-up in WT and Delta-N CLC-2. Left panels: Time course data. Following an initial voltage-family measurement (*I–V* protocol as in panel A, taken at point '1'), currents were monitored by 1 s pulses to −100 mV every 5 s for 5 min, after which a second voltage-family measurement was made (point 2), followed by application of AK-42 to facilitate leak subtraction, and a final voltage-family measurement at point 3. *I–V* traces are shown at right. (**G**) Summary data for 'run-up' experiments. Left panel: Normalized time-dependent currents for WT and Delta-N. Currents were first leak-subtracted (using the steady-state current after AK-42 application) and then normalized to the amplitude of the current measured in the first step of the 5-min sequence. Right panels: Leak-subtracted current levels at −100 mV from 'initial' and 'final' IV traces measured at points '1' and '2' in the time course (panel F). Average initial and final currents (pA ± SEM, $n = 8$) are WT: −600 ± 102 and −1400 ± 260; Delta-N: −3300 ± 630 and −2900 ± 420.

The online version of this article includes the following source data and figure supplement(s) for figure 6:

**Source data 1.** Original file for the Western blot analysis in *Figure 6D* (anti-GFP).

**Source data 2.** PDF containing *Figure 6D* and original scan of the Western blot (anti-GFP) with highlighted band and sample labels.

**Source data 3.** Excel file containing numerical data for electrophysiology and Western blot results shown in *Figure 6C–F*.

**Figure supplement 1.** Data traces from the CLC-2 run-up experiments summarized in *Figure 6G*.

**Figure supplement 2.** CLC-2 current rectification.

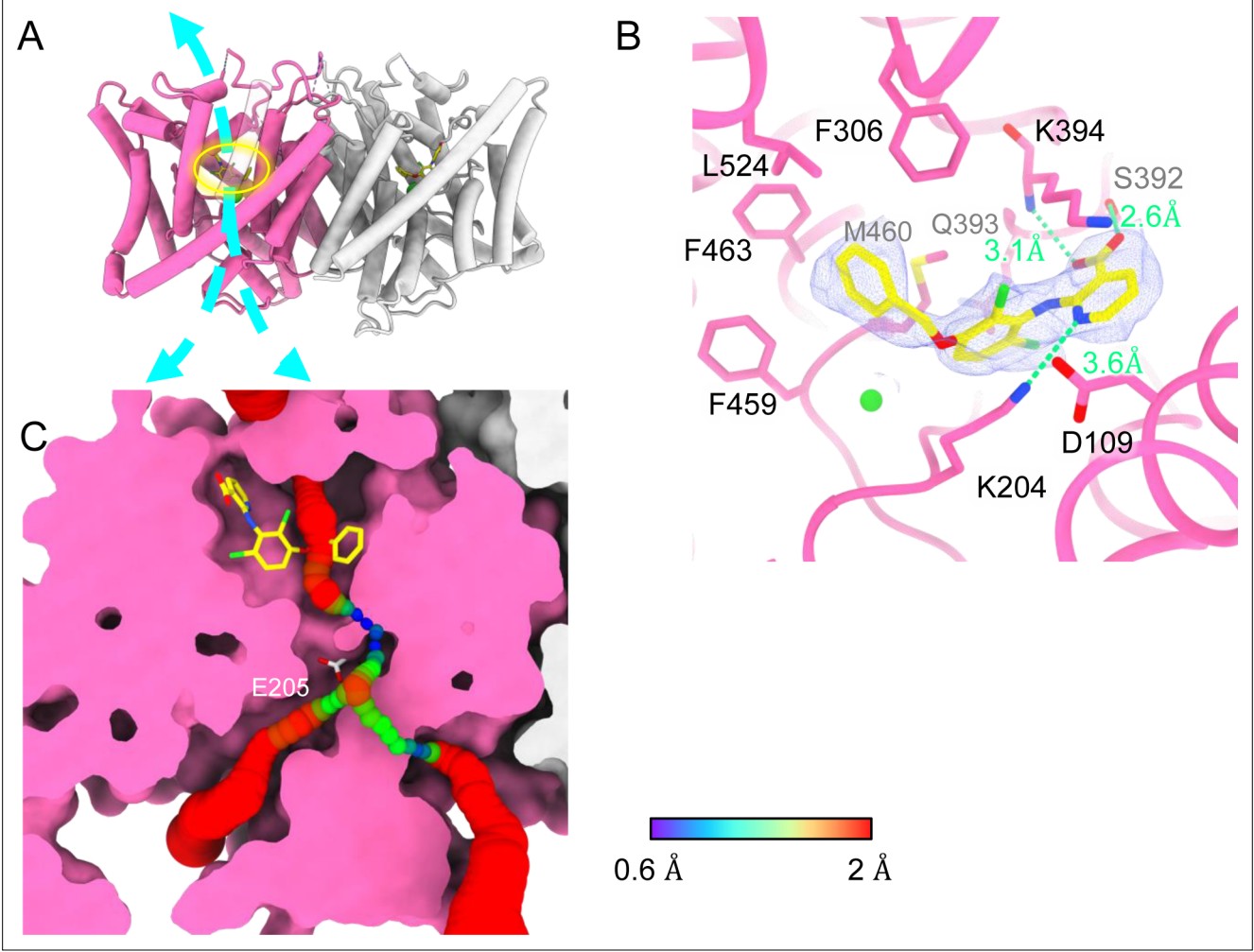

**Figure 7.** Bound AK-42 blocks the Cl⁻ pathway in CLC-2. (**A**) Transmembrane (TM) domain structure of the CLC-2 structure in complex with AK-42 (CLC2-TM-AK42) at 2.74 Å. The identical subunits of the homodimer are shown in hot pink and gray; Cl⁻ ions are shown as green balls. Bound AK-42 is highlighted with a yellow circle on the pink subunit. The cyan arrow indicates the Cl⁻ pathway. (**B**) Zoomed-in view of the AK-42-binding site (contour level: 0.91, 6.0 σ). (**C**) A sliced view of CLC-2 showing AK-42 (yellow carbon atoms; CPK coloring on non-carbon atoms) and E$_{gate}$ (CPK sticks). Both AK-42 and E$_{gate}$ obstruct the Cl⁻ pathway, displayed as a color map.

The online version of this article includes the following figure supplement(s) for figure 7:

**Figure supplement 1.** CryoEM workflow of the CLC2-TM-AK42 single-particle cryoEM data processing.

**Figure supplement 2.** Micrograph and 2D classes and structure validation of CLC2-AK42.

**Figure supplement 3.** Model validation using $Q$-scores.

**Figure supplement 4.** cryoEM map and structure of CLC2-AK42.

**Figure supplement 5.** Helix map of CLC2-TM-AK42.

**Figure supplement 6.** Density overlay at key Cl⁻ pathway residues.

**Figure supplement 7.** AK-42-binding site comparison between CLC2-TM and CLC2-TM-AK42.

**Figure supplement 8.** Structural comparison of CLC2-TM and CLC2-TM-AK42.

**Figure supplement 9.** AK-42-binding site comparison between CLC-2 and CLC-1.

*2023*); one had significantly lower energy (~17 kcal/mol) and strain on the ligand (10.2 kcal/mol), considering the ligand and nearby residues only. For the lower energy conformation, the $Q$-scores of AK-42 in the map are 0.74 and 0.76 in the two subunits, respectively. Both are higher than the expected $Q$-score at this resolution (0.64), which indicates a good fit of AK-42 to the cryoEM density (*Pintilie et al., 2020*). A similar AK-42-binding pose was observed in another recently published

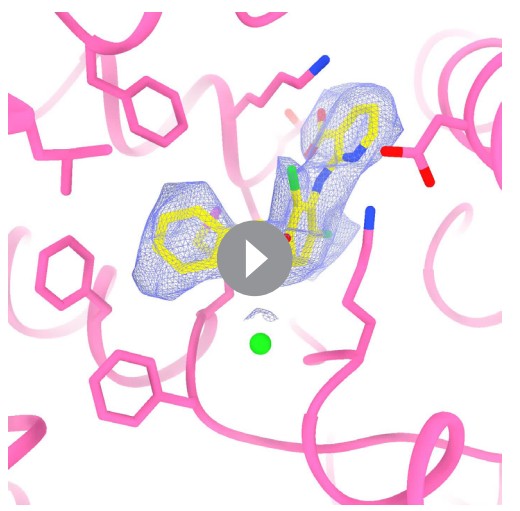

**Video 6.** AK-42 density.
https://elifesciences.org/articles/90648/figures#video6

structure of CLC-2 (*Ma et al., 2023*). The RMSD between our CLC2-TM-AK42 structure and that of Ma et al. is 0.655 Å, and the RMSD between the apo TM structures is 0.756 Å.

In the binding site, AK-42's phenyl ring nestles into a hydrophobic pocket lined by F306, F463, F459, M460, and L524 (*Figure 7B*). The carboxyl group of AK-42 forms hydrogen bonds with the side chain of S392 and backbone of K394 (*Figure 7B*). These polar interactions are consistent with mutagenesis data. At 30 nM AK-42, inhibition of S392A is reduced ~50% compared to WT, consistent with disruption of the observed hydrogen bond, while inhibition of K394A is not significantly different from WT, consistent with a backbone rather than a side-chain interaction (*Koster et al., 2020*; *Ma et al., 2023*). A third hydrogen bond – between K204 and the pyridine nitrogen of AK-42 – also contributes to AK-42's stability in the binding pocket. A pivotal role for the pyridine nitrogen was discovered during the development of AK-42, where it was found that inhibitor potency is increased by over two orders of magnitude by addition of a nitrogen at this position but not by addition of a nitrogen at any other position in the ring (*Koster et al., 2020*). The interaction of this pyridine nitrogen with K204 is supported by mutagenesis data, which shows that inhibition of K204M and K204R (K210M and K210R in rat CLC-2) is reduced ~threefold compared to WT (*Koster et al., 2020*; *Ma et al., 2023*). The distance between the nitrogen atoms on K204 and AK-42 is 3.6 Å, within acceptable hydrogen bond distance, while the distances between K204 and any other atoms on AK-42 are too far to allow hydrogen bond formation.

Notably, density corresponding to the N-terminal hairpin structure is not seen in the AK-42-bound structure. After intensive 3D classification, only very minimal hairpin density was observed in a few classes. It thus appears that the majority of AK-42-bound CLC-2 particles have a flexible N-terminus that is not blocking the Cl⁻ pathway. There is no obvious explanation for this difference compared to the apo structure. The orientation of the CTD in the AK-42-bound structure is the same as observed in the apo structure that shows N-terminal hairpin binding; the density for the J-K helix is weaker than in the apo dataset, but still sufficient to allow modeling of the helix. The TM regions also appear similar. To scrutinize for subtle conformational changes, we calculated a distance difference matrix (*Nishikawa et al., 1972*) comparing CLC2-TM and AK-42-bound CLC-2 (*Figure 7—figure supplement 8*). The minor changes observed do not readily suggest a structure-based hypothesis for the observed difference in N-terminus. It could be that AK-42 inhibits dynamics that are essential for hairpin peptide binding.

## Discussion

Our CLC-2 channel structures display quintessential CLC characteristics, including a homodimeric arrangement of subunits, TM domains composed of 18 helical segments, and conserved residues surrounding central anion-binding sites $S_{ext}$ and $S_{cen}$ (*Jentsch and Pusch, 2018*). At 2.5 Å resolution, we observe clear density for critical features, including a Cl⁻ ion at $S_{ext}$ and the $E_{gate}$ residue at $S_{cen}$ (*Figure 2C*). In this position, $E_{gate}$ blocks the pore, generating a closed conformational state – as expected for CLC-2 at 0 mV (*Figure 1*).

### AK-42-binding site and inhibition potency

A distinctive feature of AK-42 – unique among CLC inhibitors – is its high selectivity for CLC-2 versus other channels, transporters, and receptors (*Koster et al., 2020*). The $IC_{50}$ for inhibition of CLC-2 by AK-42 (17 nM) is more than four orders of magnitude lower than the $IC_{50}$ for inhibition of CLC-1,

CLC-2's closest homolog. CLC-1 and CLC-2 are 50% identical overall, and within the AK-42-binding site only five residues are different (*Figure 7—figure supplement 9*). Structural comparison of CLC-1 (PDB ID: 6coy) with CLC2-AK42 reveals that these five residues (I112 (S139), S392 (M419), Q393 (P420), K394 (R421), and L397 (I424)) exhibit only relatively minor differences in backbone conformation. While the S392 side chain contributes to a hydrogen-bond interaction with AK-42 (*Figure 7B*), which M419 in CLC-1 cannot provide, this interaction alone cannot explain the four orders of magnitude difference in AK-42 potency. Indeed, the mutation S392A reduces inhibition by 30 nM AK-42 only ~threefold (*Ma et al., 2023*). Similarly, Q393P (Q399P in rat CLC-2) reduces inhibition by less than fivefold (*Koster et al., 2020*; *Ma et al., 2023*). A steric clash between F279 (F252 in CLC-2) and the benzyl substituent in AK-42 may also contribute to the lack of potency of AK-42 toward CLC-1 (*Figure 7—figure supplement 9*). In CLC-2, this same residue (F252) is shifted away from the binding site, opening a sub-pocket that the benzyl group occupies. This difference is not simply induced by AK-42 binding, as F252 is also shifted away from the binding site in the CLC-2 apo structure. In the structure–activity relationship studies used to develop AK-42, the benzyl substituent increased selectivity approximately fivefold (*Koster et al., 2020*), thus supporting the significance of this interaction to selectivity. Unfortunately, this hypothesis cannot be tested through mutagenesis experiments due to the extremely low expression of F252 mutants (*Ma et al., 2023*).

The pathway from the extracellular solution appears sufficiently wide for AK-42 to worm its way into the binding site (*Figure 7C*). It is not yet known whether this pathway becomes wider in the CLC-2 open state. Intriguingly, the lack of density for the hairpin peptide in the AK-42-bound structure suggests that hairpin and AK-42 binding may be mutually exclusive. Studies to evaluate state dependence of AK-42 inhibition and of hairpin block will be valuable for understanding CLC-2 conformational change and gating dynamics.

## $E_{gate}$ conformation

CLC-2 is the first CLC channel homolog to be seen in a closed conformational state. While it was expected that $E_{gate}$ would localize to the $Cl^-$ permeation pathway in the closed conformational state, it was not certain whether the $S_{ext}$ or $S_{cen}$ site would be preferred. The structure of CLC-1 provides an argument that $S_{ext}$ might be preferred. In CLC-1, anion density is observed at $S_{ext}$ but not at $S_{cen}$ (*Park and MacKinnon, 2018*). It was hypothesized that the lack of density at $S_{cen}$ in CLC-1 may reflect a reduced anion-binding affinity that contributes to increased $Cl^-$ conduction (higher $Cl^-$ throughput) by the CLC channels compared to the transporters (*Park and MacKinnon, 2018*), as all structures of WT CLC transporters show anion density (either $Cl^-$ or $E_{gate}$) at both $S_{ext}$ and $S_{cen}$ (*Dutzler et al., 2003*; *Chavan et al., 2020*; *Schrecker et al., 2020*). If true for all channels, one might expect $E_{gate}$ in CLC-2 to occupy $S_{ext}$ rather than $S_{cen}$ in the closed state. However, we do not observe this occupancy, and therefore the decreased anion affinity at $S_{cen}$ is not the universal reason for increased $Cl^-$ conduction in CLC channels relative to the transporters. Instead, our CLC-2 structure is consistent with the predictions made by Arreola et al., who used homology modeling in concert with detailed electrophysiological analyses to conclude that the CLC-2 gate is formed by the $E_{gate}$–$Tyr_C$ pair at $S_{cen}$ (*De Jesús-Pérez et al., 2021*). Additional experiments, to directly measure anion affinity and to determine structures of CLC-2 and CLC-1 in both open and closed conformations are needed to better understand these channels' similarities and differences.

## CTD conformation

CLC-2 is also the first CLC in which two conformations of the CTD are observed. While many previous studies indicate that conformational change of the CTDs plays a critical role in CLC gating (*Jentsch and Pusch, 2018*; *Grieschat et al., 2020*), the mechanistic details are currently fuzzy. Unlike other CLC variants, CLC-2 can still conduct $Cl^-$ when its CTD is removed; however, the modified gating kinetics of this mutant do suggest a role of the CTD in regulating channel function (*Garcia-Olivares et al., 2008*). The impact of mutations in various regions of the CLC-2 CTD further supports its role in gating (*Yusef et al., 2006*; *Paul et al., 2007*; *Stölting et al., 2013*; *Giorgio et al., 2017*), as do the effects of ATP binding on channel kinetics (*Saint-Martin et al., 2009*; *Stölting et al., 2013*). While we do not see ATP bound in our structure (none was added to the protein preparation), the binding site region overlaps reasonably well with the structure of the ATP-bound CLC-5 CTD (*Figure 4—figure supplement 5*). The physiological significance of an intact CTD is underscored by CTD mutations and truncations

of CLC-2 that occur in human leukoencephalopathy and retinal degeneration (*Giorgio et al., 2017*; *Guo et al., 2019*; *Xu et al., 2023*). It is likely that the structural rearrangement we observe strongly influences channel gating: the CTD upward hinge rotation restricts access to the hairpin-blocking site and substantially changes CTD interactions with the TM domain.

## Ball-and-chain gating

Steric block of the Cl⁻ permeation pathway by the N-terminal hairpin peptide (*Figure 5*) rekindles the hypothesis of a 'ball-and-chain'-type gating mechanism for CLC-2, first proposed over 30 years ago (*Gründer et al., 1992*). In those studies, residue Q19 in the N-terminal hairpin, which hydrogen bonds to Q153 in the TM domain (*Figure 5C*), was identified as a residue critical to the 'ball' blocking mechanism (*Gründer et al., 1992*). In follow-up studies, it was concluded that the 'receptor' site on the TM domain is formed by residues in a stretch from 342 to 351 (*Jordt and Jentsch, 1997*) on helix J, which also forms part of the hairpin-docking site (*Figure 5C*, *Figure 5—figure supplement 6A*). The hairpin–TM interface identified in our structure also exquisitely harmonizes with results from studying human mutations associated with primary aldosteronism. These gain-of-function mutations, which map squarely onto the peptide hairpin and surrounding TM region (*Figure 5—figure supplement 6B*), cause increases in CLC-2 currents that depolarize the glomerulosal membrane potential thus leading to disease (*Fernandes-Rosa et al., 2018*; *Scholl et al., 2018*).

Our electrophysiology results also support a ball-and-chain gating mechanism. First, we demonstrate that the increase in current in hairpin-deleted channels reflects an increase in channel activity rather than an increase in surface expression (*Figure 6D, E*), consistent with loss of a channel-closing mechanism. Second, we show that WT but not Delta-N CLC-2 currents increase over time (*Figure 6F, G*), consistent with relief of the hairpin block. Importantly, our use of the selective inhibitor AK-42 (*Koster et al., 2020*) provides confidence that the current increase is specific to CLC-2 and not a result of leak or a run-up of other Cl⁻ currents, which notoriously lurk in all mammalian cells and can confound interpretation of experimental results. In the context of our structure and the mutagenesis studies discussed above, our electrophysiological results fortify a ball-and-chain model for CLC-2 gating.

## Ball-and-chain gating as a potential mechanism for CLC-2 current variability

How do we explain the ostensible discrepancy between our results, which support a ball-and-chain model, and the previously published results that were used to argue against a ball-and-chain type gating mechanism? We think the discrepancy is not as dire as it originally seemed. The previous discrepancy is as follows. In studies of an N-terminal CLC-2 deletion mutant in HEK 293 cells, Varela et al. observed currents that differed depending on whether whole-cell or nystatin-perforated patch-clamp recording was used: with whole-cell recording, deletion-mutant currents displayed rectification similar to WT CLC-2 but with much faster kinetics; with perforated patch recording, currents were voltage independent and non-rectifying (*Varela et al., 2002*). At the time, this difference was interpreted to mean that the open-channel phenotype (and therefore ball-and-chain gating) was artifactual, due to a diffusible intracellular factor binding to the N-terminal deletion channel, but not the WT channel, locking it open in the perforated patch recording configuration (but diffusing away in whole-cell recording experiments). However, Varela et al. also aptly pointed out that mutagenesis experiments did support a role for the CLC-2 N-terminus in channel gating. Furthermore, they noted that the different results between their two recording configurations could be due to differences of physiological relevance, such as osmotic forces and/or channel–cytoskeleton interactions (*Gründer et al., 1992*; *Ahmed et al., 2000*). Such factors could also explain the differences between our whole-cell recordings and those of Varela et al. We note that our Delta-N currents observed with whole-cell recording (*Figure 6—figure supplement 1*) are similar to currents displayed by the aldosteronism peptide-hairpin mutant G24D in perforated patch-clamp recordings (*Göppner et al., 2019*), supporting that our use of whole-cell recording is not introducing an artifact.

We hypothesize that sensitivity of ball-and-chain gating to physiological variables may underlie the tremendous variation in kinetics and voltage dependence observed among recordings of natively expressed CLC-2 (*Bösl et al., 2001*; *Jentsch and Pusch, 2018*). Such variation could arise from differences in one or more of the factors known to modulate CLC-2 gating – osmotic forces, TM voltage, ATP, phosphorylation, or interactions with the cytoskeleton or other cellular proteins (*Nighot et al.,*

*2017*; *Jentsch and Pusch, 2018*; *Okamoto et al., 2019*) – and could be further modulated by the expression of N-terminal splice variants (*Furukawa et al., 1995*; *Cid et al., 2000*). We propose that mechanistic studies of the CLC-2 gating run-up we report here (*Figure 6F*) will be valuable to guide studies on the variability of CLC-2 currents in native systems and how it relates to ball-and-chain gating.

## Revised framework for CLC-2 gating

The established framework for CLC channel gating (*Jentsch and Pusch, 2018*) is based on pioneering investigations on the *Torpedo* homolog CLC-0. These investigations identified two separate mechanisms responsible for channel gating (*Miller, 1982*). The first gating mechanism, referred to as 'fast' or 'protopore' gating, opens and closes each pore of the homodimer independently on a millisecond timescale. $E_{gate}$ (*Figure 1*) plays a central role in this protopore gating (*Dutzler et al., 2003*), with voltage dependence arising from interactions between $E_{gate}$ and ions in the pore (*Pusch et al., 1995*). The second gating mechanism, referred to as 'slow' or 'common' gating, is orders of magnitude slower (seconds to tens-of-seconds timescale) than 'fast' gating and operates in a highly cooperative manner to open and close both CLC-0 pores simultaneously. The simultaneous opening/closing of pores that are separated by >30 Å, together with the exceptionally high-temperature dependence of slow gating (*Pusch et al., 1997*), suggests that large structural rearrangements are involved. Indeed, large movements of the CTDs were observed in a fluorescence resonance energy transfer study of CLC-0 slow gating (*Bykova et al., 2006*).

In CLC-2, electrophysiological studies of WT and mutant channels were used to illustrate that CLC-2 has two gating mechanisms (*Zúñiga et al., 2004*; *de Santiago et al., 2005*; *Yusef et al., 2006*). However, it remains uncertain as to whether these two mechanisms are analogous to the fast protopore and slow common gating mechanisms observed in CLC-0. On the one hand, studies demonstrating that interactions between $E_{gate}$ and ions in the pore play a critical role in gating support that CLC-2 has a 'fast' protopore gating mechanism (*Niemeyer et al., 2003*; *Niemeyer et al., 2009*; *Sánchez-Rodríguez et al., 2010*; *Sánchez-Rodríguez et al., 2012*; *De Jesús-Pérez et al., 2016*). On the other hand, there are substantial differences between CLC-2's 'slow' gating mode and CLC-0's common gating in terms of kinetics and temperature dependence. Moreover, unlike CLC-0 (*Lin et al., 1999*), CLC-2 slow and fast gating cannot be separately affected by mutation (*Zúñiga et al., 2004*; *de Santiago et al., 2005*; *Yusef et al., 2006*; *De Jesús-Pérez et al., 2021*). Finally, in contrast to CLC-0, it has not been demonstrated that the 'slow' gate in CLC-2 corresponds to a common gating process operating on both pores simultaneously.

Our results strongly motivate a revised framework for understanding CLC-2 protopore gating. At least two events are needed to open each protopore – relief of block by the N-terminal hairpin (ball-and-chain gating) and $E_{gate}$ movement (*Figure 8*). A key aspect of our revised framework is that

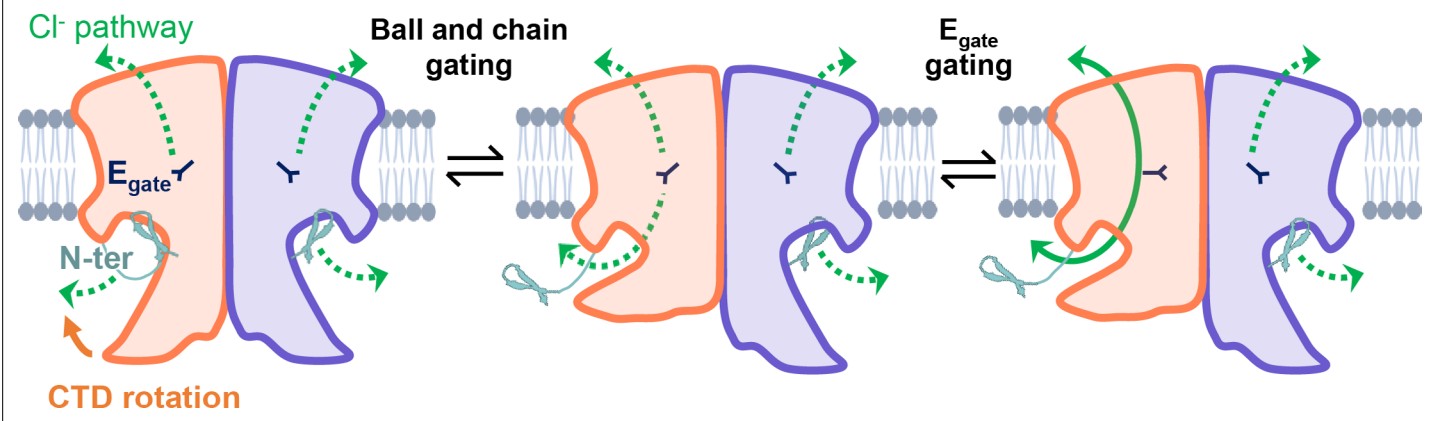

**Figure 8.** CLC-2's gating is illustrated based on our new structural data. Opening of each protopore requires at least two events – unblock by the N-terminal hairpin (N-ter) and rotation of $E_{gate}$ away from the permeation pathway. The dashed green arrow represents the Cl⁻ conduction pathway. Hairpin unblock (ball-and-chain gating) is coupled to conformational change of the C-terminal domain (CTD): rotation of the CTD toward the membrane precludes hairpin binding, thus facilitating channel opening. Opening of the channel pore following these two events is depicted for one subunit of the homodimer (filled green arrow).

ball-and-chain gating is coupled to CTD conformational change: close examination of the state with the CTD rotated (*Figure 4A*) reveals that the hairpin cannot access its pore-blocking site in this state. Thus, CTD rotation facilitates hairpin release and channel opening. This structural model is consistent with electrophysiological studies showing that removal of the CTD speeds CLC-2 activation and deactivation kinetics (*Garcia-Olivares et al., 2008*). In contrast to CLC-0, where movements in the CTD have been linked to the highly cooperative conformational change involved in common gating (*Estévez et al., 2004*; *Bykova et al., 2006*), our structures illustrate that in CLC-2 the CTDs can move independently of one another.

CLC-2 gating is more complex than depicted in our simple cartoon, and many questions remain. First, prior results suggest that CLC-2 'slow' gating involves conformational change beyond the $E_{gate}$ and hairpin/CTD motions depicted in our model. Even when $E_{gate}$ is neutralized, CLC-2 retains some voltage-dependent gating (*Niemeyer et al., 2003*), indicating a conformational change beyond $E_{gate}$. This additional voltage-dependent gating cannot be solely due to N-terminal ball-and-chain gating, since channels with N-terminal deletions retain biexponential voltage dependence as is seen in the wild-type channels (*Varela et al., 2002*). Second, electrophysiological studies indicate that CLC-2 has at least two open states (*de Santiago et al., 2005*), but the physical basis for these different open states is unknown. These points strongly motivate studies to elucidate additional conformational changes in CLC-2. Whether such changes include a highly cooperative common gating mechanism will require functional studies to complement structures, ideally with visualization of gating in single-channel recordings. Augmenting complexity in understanding CLC-2 is our observation that channel rectification changes upon N-terminal deletion or ball-and-chain gating (*Figure 6F*; *Figure 6—figure supplement 2*): the molecular underpinnings of this change in rectification are wholly unknown. Addressing all these questions will be important for understanding CLC-2's diverse functional characteristics and physiological roles in various tissues and cells (*Bösl et al., 2001*; *Jentsch and Pusch, 2018*).

## Summary

The structures and electrophysiological data presented here reveal a ball-and-chain mechanism involved in CLC-2 gating and offer a framework for understanding CLC-2's multiple gating mechanisms. In addition, the structures illuminate details for molecular recognition by the selective inhibitor AK-42, the first and currently only known small-molecule modulator that is both potent and selective for a CLC family member.

## Materials and methods
### Molecular biology and AK-42 synthesis
Full-length human CLC-2 (Uniprot ID: P51788) was fused to a C-terminal twin-strep tag with a linker of 5'- TCGAATTCT<u>TTGGAAGTTTTGTTTCAAGGTCCA</u>ACTGCTGCCGCCGCT-3' containing a HRV 3C protease cleavage sequence (underlined). This human C-terminal twin-strep-tagged CLC-2 fusion construct was cloned into a modified pFastBac vector that contains a CMV promoter for protein expression in mammalian cells (*Goehring et al., 2014*; *Park et al., 2017*). For electrophysiology and cell surface protein biotinylation/isolation experiments, the twin-strep tag was replaced by an EGFP tag. A C-terminal GFP tag was previously shown to have no significant effect on CLC-2 channel kinetics or voltage dependence (*Park et al., 1998*). In the N-terminal deletion CLC-2 (Delta-N), residues 14–27 (ALQYEQTLMYGRYT) were removed from full-length (WT) CLC-2.

AK-42 was synthesized and quantified as previously described (*Koster et al., 2020*).

### Protein expression/ purification
The full-length CLC-2 construct was transformed to DH10Bac competent cells (Invitrogen) to isolate baculovirus bacmid. The bacmid was transfected to sf9 cells using Cellfectin-II (Invitrogen) to produce baculovirus. The baculovirus were then further amplified twice in sf9 cells. The protein was expressed with HEK293 GnTI⁻ cells with the amplified baculovirus. HEK293 GnTI⁻ cells were cultured in Freestyle 293 medium (Invitrogen) supplemented with 2% fetal bovine serum on a shaker at 37°C in the presence of 8% $CO_2$ to a density of $3 \times 10^6$ cells per ml, then infected with 5% vol/vol baculovirus.

After culturing for another 8–16 hr, sodium butyrate was added at final concentration of 10 mM, then further expressed for 48 hr at 37°C before harvest. Cells were pelleted and stored at −80°C.

Frozen cell pellets were resuspended in resuspension buffer containing 50 mM 4-(2-hydroxyethyl)-1-piperazineethanesulfonic acid (HEPES), pH 7.5, 300 mM NaCl, 1 mM dithiothreitol (DTT), 1 mM ethylenediaminetetraacetic acid (EDTA), and 1 protease inhibitor cocktail tablet (MedChem Express), then briefly lysis cells with dounce homogenizer for 10–15 times. Cellular debris were collected by centrifugation at 18k rpm for 30 min at 4°C, and then resuspended with resuspension buffer supplemented with 1% lauryl maltose neopentyl glycol (LMNG) and 0.1% cholesteryl semisuccinate (CHS). After extraction for 2 hr, the lysate was centrifuged at 18k rpm for 40 min at 4°C. The clarified lysate was incubated with 1 ml of strep-Tacin XT high-capacity resin (IBA) for 2 hr at 4°C. Resin was washed with wash buffer containing 50 mM HEPES, pH 7.5, 300 mM NaCl, 1 mM DTT, 1 mM EDTA, 0.02% LMNG, and 0.002% CHS. Purified protein was released from resin with 50 µg HRV 3C protease and incubated at 4°C for overnight. The retrieved protein was concentrated to 0.5 ml with Amicon Ultra (50 kDa cutoff, EMD Millipore) and followed by size-exclusion chromatography (SEC) using a superdex 200 Increased 10/300 chromatography column on an AKTA Purifier 10 FPLC system (GE Healthcare) with buffer containing 20 mM HEPES, 150 mM NaCl, 1 mM DTT, 0.5 mM EDTA, 0.002% LMNG, and 0.0002% CHS. Protein fractions were pooled, concentrated with Amicon Ultra (50 kDa cutoff, EMD Millipore) to ~5 mg/ml, and immediately used for cryoEM grid preparation.

For CLC-2 and AK-42 complex, AK-42 was first dissolved in dimethyl sulfoxide (DMSO) to 45 mM, then further diluted with CLC-2 SEC buffer to 2 mM and mixed with purified and concentrated CLC-2 protein (2.9 mg/ml) to make final solution with CLC-2/AK-42 at a molar ratio of 1:10 (20 µM CLC-2 with 200 µM AK-42). The mixture was incubated at 4°C for 3 hr and then used for cryoEM grid preparation.

## CryoEM data collection and processing

Three microliters of the purified CLC-2 or CLC-2/AK-42 mixture was applied to glow-discharged copper Quantifoil R1.2/1.3 or R2/1 holey carbon grids (Quantifoil). Grids were incubated for 15 s, then blotted for 4 or 6 s at 100% humidity, then plunge-frozen in liquid-nitrogen-cooled liquid ethane using a Vitrobot Mark IV (Thermo Fisher Scientific) at 4°C.

The grids were imaged on a Titan Krios electron microscope (Thermo Fisher) operated at 300 kV with a Falcon 4 direct electron detector (Thermo Fisher Scientific) and SelectrisX energy filter (Thermo Fisher Scientific). Micrographs were recorded at a magnification of ×130,000 (pixel size of 0.946 Å per pixel) with EPU software (Thermo Fisher Scientific). Each image was composed of 40 individual frames with total cumulative dose of 50 e⁻ per Å². For CLC-2 dataset, images were collected with an exposure time of 7.5 s and a dose rate of 6.7 e⁻ per second per Å² (OR 6.0 e⁻ per second per pixel). For CLC-2/AK-42 dataset, images were collected with an exposure time of 5.6 s and a dose rate of 8.9 e⁻ per second per Å² (OR 8.0 e⁻ per second per pixel). Defocus values were set from −1.0 to −2.0 µm.

CryoEM data processing was performed mainly using CryoSPARC 3.2 (Structura Biotechnology Inc), except those mentioned specifically. For CLC-2 sample, a total of 14,198 movie stacks were subjected to motion correction. The contrast-transfer function (CTF) parameters were estimated using patch CTF. Initially, a total of 56,091 particles were auto-picked and subjected to reference-free 2D classification. A total of 7 class averages were selected as templated for automated particle picking. A total of 4,999,780 particles were auto-picked and extracted with a box size of 256 pixels. The extracted particles were then subjected to 2D classification to remove ice, contaminants, and aggregates. The following ab initio reconstruction and 3D refinement (for all structures presented in this paper) are all performed with C1 symmetry (no symmetry imposed). The cleared-up classes were used for ab initio reconstruction with two classes. These models were used as reference for a heterogenous refinement with no symmetry imposed. The best class from heterogenous refinement which contained 2,415,222 particles were subjected to non-uniform 3D refinements, followed by local 3D refinements, yielding final CLC2-TM map at 2.46 Å, as determined by the gold standard measure of Fourier shell correlation (FSC) using a cutoff of 0.143. Local resolution was estimated with relion 3.1. To further improve density map of CTD, the selected 2,415,222 particles were divided into 10 subsets for 3D classification in relion 3.1. A reference map for 3D classification was generated by the 3D ab initio model in CryoSparc. Two of the classes, containing 56,580 and 39,272 particles, respectively, showed distinct CTD features and were used for further non-uniform 3D refinement and local 3D refinements, separately. The resolution was determined by FSC using a cutoff of 0.143 at 2.75 and 2.76 Å, respectively.

Processing for CLC2-AK42 sample was executed similarly as described above. In brief, a total of 14,300 movie stacks were motion corrected by CryoSPARC. After auto-picking, 5,214,695 particles were extracted and subjected to 2D classification. After 2D classification, 2,391,813 particles were used for non-uniform 3D and local refinement to generate the final 2.74 Å map. To further improve the CTD density map, 3D classification was performed. The best of the 10 classes, containing 66,251 particles, was used for further refinement. Local refinement with masking of the CTD allowed a resolution 4.03 Å to be reached.

## Model building and refinement

The initial model of CLC-2 was generated by SWISS-MODEL using human CLC-1 (PDB:6qvc) as a homology model. It was fit to each map and manually rebuilt using Coot (*Emsley and Cowtan, 2004*) (v0.91). The models were further refined with ISOLDE (*Croll, 2018*) in ChimeraX (*Pettersen et al., 2004*) (v.1.6) and Phenix (*Liebschner et al., 2019*) (v1.20) real space refinement. The two potential conformations for the AK-42 ligand were tested using the QMR method (*Liebschner et al., 2023*). *Q*-scores were performed with the MapQ Chimera plugin v1.9.12.

## Cell surface protein biotinylation and isolation

CHO K1 cells (ATCC CCL-61) were cultured at 37°C [95/5% $O_2/CO_2$ ratio] in F12K media (ATCC, Catalog no. 30-2004) supplemented with 10% fetal bovine serum (Gibco) and 1% penicillin/streptomycin (Gibco). Cells were seeded at $5 \times 10^6$ in 15 cm dishes. After 48 hr, the confluency reached around 60%, and cells were transfected with 6 µg of CLC-2 plasmid, either WT or Delta-N, using Lipofectamine LTX, opti-MEM, and PLUS reagent (Invitrogen). For the control sample, CHO cells were seeded at the same time as the transfected groups and kept culturing in parallel. 48–72 hr after transient transfection, two 15 cm dishes cells for each sample (WT, Delta-N, or control) were combined to create one sample for use in the biotinylation of surface proteins assay, which was performed using a Pierce cell surface protein biotinylation and isolation kit (Thermo Scientific, A44390). Briefly, cells were washed twice in phosphate-buffered saline (PBS) (0.1 M sodium phosphate, 0.15 M NaCl, pH 7.2) and then incubated with PBS containing 0.25 mg/ml of sulfosuccinimidyl 2-(biotin-amido)-ethyl-1,3-dithiopropionate (EZ-link Sulfo-NHS-SS-biotin, Pierce) at room temperature for 10 min. Cells were incubated in ice-cold buffer (25 mM Tris, 0.15 M NaCl, pH 7.2) twice to quench the labeling. Cells were then scraped and lysed in lysis buffer from the Kit. Samples were incubated at 4°C for 30 min and then centrifugated at 15,000 × *g* for 5 min. Following centrifugation, the extracted membrane samples from the supernatant were incubated with 50 µl NeutrAvidin Agarose (Pierce) at 4°C for 2 hr, then washed with wash buffer from the kit four times. Finally, the isolated proteins were eluted with 60 µl elution buffer (from the kit) containing 100 mM DTT. 40 µl of each sample was used for Western blotting detection with 1:2,500 anti-GFP antibody (Invitrogen A11122) and 1:10,000 anti-rabbit secondary antibody (Licor 926-68073). The relative expression level was calculated by quantifying the intensities of the 100-kD bands on the Western blot, using a Li-Cor Odyssey 9120 Infrared Imaging System. The samples from three independent experiments (independent transfections on different days) were run on a single Western blot, shown in *Figure 6D* and in *Figure 6—source data 1 and 2*.

## Cell culture and patch-clamp recording

CHO K1 cells (ATCC CCL-61) were cultured at 37°C [95/5% $O_2/CO_2$ ratio] in F12K media (ATCC, Catalog no. 30-2004) supplemented with 10% fetal bovine serum (Gibco) and 1% penicillin/streptomycin (Gibco). At 60% confluency, cells were transfected with 120 ng CLC-2, either WT or Delta-N, using Lipofectamine LTX, opti-MEM, and PLUS reagent (Invitrogen). 105 ng of a soluble GFP construct was co-transfected with the CLC-2 construct to improve visualization of transfected cells.

Whole-cell patch-clamp experiments were performed using an Axopatch 200B amplifier attached to a Digidata 1440A. Data were acquired at 10 kHz with filtering at 5 kHz. Traces shown in figures were filtered offline in Clampfit at 1 kHz (8-pole Bessel). Borosilicate glass micropipettes (Sutter Instruments BF150-86-10) were pulled and fire polished to a tip diameter with a resistance of 1.8–3.4 MΩ. For whole-cell patch-clamp recordings, the external solution was composed of 148 mM CsCl, 2 mM $CaCl_2$ (from a 1-M aqueous stock), 100 mM D-mannitol, and 10 mM HEPES, adjusted to pH 7.4 with 2 M CsOH. The internal solution was composed of 146 mM CsCl, 5 mM ethylene glycol-bis(β-aminoethyl ether)-*N,N,N',N'*-tetraacetic acid (EGTA), 5 mM NaF, 60 mM D-mannitol, and 10mM HEPES, adjusted to pH 7.4 with 2 M CsOH. Bath perfusion of external solution (2 ml/min) was initiated after whole

cell and series resistance compensation (70% or more) was established. For leak subtraction, 300 nM AK-42 was perfused at 2 ml/min. AK-42 was diluted from a 1-mM stock solution in DMSO (0.03% final [DMSO]).

## MD simulations

### System setup for MD simulations

The apo dimer structure was used for simulations. For all simulations, hydrogen atoms were added, and protein chain termini were capped with neutral acetyl and methylamide groups. Titratable residues were kept in their dominant protonation state at pH 7. Histidine residues were modeled as neutral, with a hydrogen atom bound to epsilon nitrogen. The Dowser program was used to hydrate pockets within and around each structure (*Morozenko and Stuchebrukhov, 2016*). Then the receptor was inserted into a pre-equilibrated palmitoyl-oleoyl-phosphatidylcholine bilayer using Dabble (*Betz, 2017*). Sodium and chloride ions were added to neutralize each system at a concentration of 150 mM. Approximate system dimensions were 131 Å × 95 Å × 125 Å.

### Simulation protocols

All simulations were run on a single Graphical Processing Unit (GPU) using the Amber18 Compute Unified Device Architecture (CUDA) version of particle-mesh Ewald molecular dynamics (PMEMD) (*Salomon-Ferrer et al., 2013*). We used the CHARMM36m parameter set for protein molecules, lipids, and ions, and the CHARMM TIP3P water model for waters (*Huang and MacKerell, 2013*). Heating (to 310 K over 137.5 ps) and equilibration (28 ns with restraints on protein and ligand) steps were performed before production simulations as described previously (*Suomivuori et al., 2020*). Trajectory snapshots were saved every 200 ps. All simulations were at least 2 μs in length.

### Simulation analysis protocols

The AmberTools18 CPPTRAJ package (*Roe and Cheatham, 2013*) was used to reimage trajectories, while Visual Molecular Dynamics (VMD) (*Humphrey et al., 1996*), PyMol (Schrodinger), and Matplotlib Python package were used for visualization and analysis. For analysis, the N-terminus peptide backbone was defined as segment name P5, excluding residues 14 15 27 28. For the CTD analysis, we used segment names P9 P3 P4 P8 excluding residues 419–578. For bar plots, individual dots show the average value in each independent simulation. Bar values show the average value across the independent simulations. The error bars for simulation results show the 68% confidence intervals of the mean (appropriate for non-parametric distributions) calculated using bootstrapping with the Seaborn Python library.

## Acknowledgements

We thank Huawei Zhang for sharing pdb coordinates and cryoEM density for CLC-2/AK-42 in advance of release. We thank Anna Koster, Bill Kobertz, Dan Collins, Jürgen Kreiter, Martin Prieto, Shwetha Srinivasan, and Chris Miller for comments on the manuscript. We also thank Nigel Moriarty (Phenix) for performing Quantum Mechanical Restraints (QMR) analysis of the AK-42 ligand. This research was funded by NIH R21NS125767 (MM and WC) and NIH R01NS113611 (MM, JDB, and ROD). CryoEM was performed at the Stanford-SLAC Cryo-EM Center ($S^2C^2$), which is supported by the National Institutes of Health Common Fund Transformative High-Resolution Cryo-Electron Microscopy program (U24 GM129541). MX was supported by a Wu Tsai Neurosciences Institute Interdisciplinary Scholar Award. SDM is supported by a Stanford Bio-X Bowes Fellowship.

## Additional information

### Competing interests

Merritt Maduke: Senior editor, eLife. The other authors declare that no competing interests exist.

## Funding

| Funder | Grant reference number | Author |
|---|---|---|
| National Institutes of Health | R21NS125767 | Merritt Maduke<br>Wah Chiu |
| National Institutes of Health | R01NS113611 | Merritt Maduke<br>J Du Bois<br>Ron O Dror |
| Wu Tsai Neurosciences Institute | Interdisciplinary Scholar Award | Mengyuan Xu |
| Stanford Bio-X | Bowes Fellowship | Steven D Miller |

The funders had no role in study design, data collection, and interpretation, or the decision to submit the work for publication.

## Author contributions

Mengyuan Xu, Conceived and carried out experiments, performed molecular biology to generate constructs, purified protein, collected and analyzed cryoEM data, built and refined experimental model, performed surface biotinylation experiments, and generated figures, Writing – review and editing; Torben Neelands, Conceived, carried out, and analyzed electrophysiology experiments, Writing – review and editing; Alexander S Powers, Conceived, carried out, and analyzed MD simulations, Writing – review and editing; Yan Liu, CryoEM data collection and analysis; Steven D Miller, Synthesized AK-42, performed transfections and cell culture for electrophysiology experiments, analyzed electrophysiology data, Writing – review and editing; Grigore D Pintilie, Model refinement, analysis, validation and deposition, writing related to ligand modeling and validation; J Du Bois, Ron O Dror, Wah Chiu, Supervision, Funding acquisition, Writing – review and editing; Merritt Maduke, Conceptualization, Funding acquisition, Supervision, Writing – original draft, Writing – review and editing

## Author ORCIDs

Mengyuan Xu ⬚ https://orcid.org/0000-0002-1734-4163
Ron O Dror ⬚ https://orcid.org/0000-0002-6418-2793
Wah Chiu ⬚ https://orcid.org/0000-0002-8910-3078
Merritt Maduke ⬚ https://orcid.org/0000-0001-7787-306X

Reviewer #1 (Public Review): https://doi.org/10.7554/eLife.90648.3.sa1
Reviewer #2 (Public Review): https://doi.org/10.7554/eLife.90648.3.sa2
Reviewer #3 (Public Review): https://doi.org/10.7554/eLife.90648.3.sa3
Author Response https://doi.org/10.7554/eLife.90648.3.sa4

# Additional files

## Supplementary files

- MDAR checklist
- Supplementary file 1. cryoEM statistics.

## Data availability

CryoEM density maps and structure coordinates have been deposited in the Electron Microscopy Data Bank (EMDB) and the Protein Data Bank (PDB), with accession codes EMD-41127 and 8TA3 for the CLC2-TM, EMD-41128 and 8TA4 for the CLC2-CTDsym, EMD-41129 and 8TA5 for CLC2-CTDasym, EMD-41126 and 8TA2 for CLC2-TM-AK42, and EMD-41130 and 8TA6 for CLC2-AK42-CTD-only. Particle stacks are deposited to the Electron Microscopy Public Image Archive (EMPIAR), with accession codes EMPIAR-11864 (CLC-2/AK-42) and EMPIAR-11865 (CLC-2). Raw micrographs are available upon request by contacting the corresponding authors; they were not deposited to EMPIAR due to large size and quota limitations. Simulation files are available at Zenodo. Source data are provided for *Figure 6*.

The following datasets were generated:

| Author(s) | Year | Dataset title | Dataset URL | Database and Identifier |
|---|---|---|---|---|
| Chiu W, Dror RO, Du Bois J, Liu Y, Maduke M, Miller S, Neelands T, Pintilie G, Powers AS, Xu M | 2023 | Cryo-EM structure of the human CLC-2 chloride channel transmembrane domain Apo state with resolved N-terminal hairpin | https://www.ebi.ac.uk/emdb/EMD-41127 | Electron Microscopy Data Bank, EMD-41127 |
| Xu M, Neelands T, Powers AS, Liu Y, Miller S, Pintilie G, Du Bois J, Dror RO, Chiu W, Maduke M | 2023 | Cryo-EM structure of the human CLC-2 chloride channel transmembrane domain Apo state with resolved N-terminal hairpin | https://www.rcsb.org/structure/8TA3 | RCSB Protein Data Bank, 8TA3 |
| Chiu W, Dror RO, Du Bois J, Liu Y, Maduke M, Miller S, Neelands T, Pintilie G, Powers AS, Xu M | 2023 | Cryo-EM structure of the human CLC-2 chloride channel transmembrane domain with symmetric C-terminal | https://www.ebi.ac.uk/emdb/EMD-41128 | Electron Microscopy Data Bank, EMD-41128 |
| Xu M, Neelands T, Powers AS, Liu Y, Miller S, Pintilie G, Du Bois J, Dror RO, Chiu W, Maduke M | 2023 | Cryo-EM structure of the human CLC-2 chloride channel transmembrane domain with symmetric C-terminal | https://www.rcsb.org/structure/8TA4 | RCSB Protein Data Bank, 8TA4 |
| Chiu W, Dror RO, Du Bois J, Liu Y, Maduke M, Miller S, Neelands T, Pintilie G, Powers AS, Xu M | 2023 | Cryo-EM structure of the human CLC-2 chloride channel transmembrane domain with asymmetric C-terminal | https://www.ebi.ac.uk/emdb/EMD-41129 | Electron Microscopy Data Bank, EMD-41129 |
| Xu M, Neelands T, Powers AS, Liu Y, Miller S, Pintilie G, Du Bois J, Dror RO, Chiu W, Maduke M | 2023 | Cryo-EM structure of the human CLC-2 chloride channel transmembrane domain with asymmetric C-terminal | https://www.rcsb.org/structure/8TA5 | RCSB Protein Data Bank, 8TA5 |
| Chiu W, Dror RO, Du Bois J, Liu Y, Maduke M, Miller S, Neelands T, Pintilie G, Powers AS, Xu M | 2023 | Cryo-EM structure of the human CLC-2 chloride channel transmembrane domain with bound inhibitor AK-42 | https://www.ebi.ac.uk/emdb/EMD-41126 | Electron Microscopy Data Bank, EMD-41126 |
| Xu M, Neelands T, Powers AS, Liu Y, Miller S, Pintilie G, Du Bois J, Dror RO, Chiu W, Maduke M | 2023 | Cryo-EM structure of the human CLC-2 chloride channel transmembrane domain with bound inhibitor AK-42 | https://www.rcsb.org/structure/8TA2 | RCSB Protein Data Bank, 8TA2 |
| Chiu W, Dror RO, Du Bois J, Liu Y, Maduke M, Miller S, Neelands T, Pintilie G, Powers AS, Xu M | 2023 | Cryo-EM structure of the human CLC-2 chloride channel C-terminal domain | https://www.ebi.ac.uk/emdb/EMD-41130 | Electron Microscopy Data Bank, EMD-41130 |
| Xu M, Neelands T, Powers AS, Liu Y, Miller S, Pintilie G, Du Bois J, Dror RO, Chiu W, Maduke M | 2023 | Cryo-EM structure of the human CLC-2 chloride channel C-terminal domain | https://www.rcsb.org/structure/8TA6 | RCSB Protein Data Bank, 8TA6 |
| Xu M, Pintilie G, Liu Y, Chiu W, Maduke M | 2024 | CryoEM structures of the human CLC-2-AK42 voltage gated chloride channel reveal a ball and chain gating mechanism | https://www.ebi.ac.uk/empiar/EMPIAR-11864 | Electron Microscopy Public Image Archive, EMPIAR-11864 |

*Continued on next page*

*Continued*

| Author(s) | Year | Dataset title | Dataset URL | Database and Identifier |
|---|---|---|---|---|
| Xu M, Pintilie G, Liu Y, Chiu Y, Maduke M | 2024 | CryoEM structures of the human CLC-2 voltage gated chloride channel reveal a ball and chain gating mechanism | https://www.ebi.ac.uk/empiar/EMPIAR-11865 | Electron Microscopy Public Image Archive, EMPIAR-11865 |
| Powers A | 2023 | MD data for "CryoEM structures of the human CLC-2 voltage-gated chloride channel reveal a ball-and-chain gating mechanism" | https://doi.org/10.5281/zenodo.10397869 | Zenodo, 10.5281/zenodo.10397869 |

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
