## [Editor Report · eLife assessment]

This study examines the human voltage-gated chloride channel CLC-2. A combination of cryo-EM, electrophysiology, and computational analysis provides **compelling** support for a "ball and chain" mechanism of inactivation. This and other findings regarding the gating and inhibition mechanisms of the channel are of **fundamental** interest to ion channel physiologists.

---

## [Referee Report · Reviewer #1 (Public Review)]

This manuscript deftly combines cryo-EM and electrophysiology to investigate gating mechanisms of human CLC-2. Although another structure of CLC-2 was recently reported, this is the first structure to report density for the absolutely critical gating glutamate, and - an even more exciting result - the first structure to identify the N-terminal gating peptide that is the heart of this manuscript. There has been previous controversy over such a gating peptide in CLC-2, but the combined structural/functional approach appears to establish a role for this peptide in gating, and sets up future experiments to understand why its effects might change under different physiological scenarios. The experiments reported here are thoughtful and well-controlled and the data presentation is excellent. For the electrophysiology experiments, the use of inhibitor AK-42 (developed by the current senior author's lab) to establish a zero current level is a welcome advance and should become standard for electrophysiological studies of CLC-2.

---

## [Referee Report · Reviewer #2 (Public Review)]

This paper makes important and novel advances that significantly enhance our understanding of the ClC-2 channel. The EM data are of high quality, and the most important argument, concerning the role of the N-terminus of the protein as an occluding inactivation gate, is very well supported by both structural, computational, and functional data (some of which is previously published). The proposal that the "run up" observed in patch clamp experiments represents relief of inactivation is interesting and compelling. The model predicts that mutations at the hairpin binding site should influence this "run up", which should be tested in the near future. Finally, the confirmation of the AK-42 binding site further solidifies evidence that this is a pore-blocking compound; the authors' argument about determinants of specificity is convincing.

---

## [Referee Report · Reviewer #3 (Public Review)]

Summary

CLC-2 channels play an important role in cellular homeostasis and electrical excitability, and dysfunctions are associated with aldosteronism and leukodystrophy. Structural insights into the functioning of CLC-2 are just emerging. CLC-2 channels are distinct among the members of the CLC family in that they are activated by hyperpolarization. Earlier studies have implicated channel regulation by a "ball-and-chain" type of channel block mechanism which underlies its strong rectification and use-dependent "run-up" properties. Structural insights into these mechanisms are currently lacking. In this manuscript, Xu et al present CryoEM structures of CLC-2 in the apo and inhibitor-bound conformations in the 2.5-2.7 A resolution range. Several novel structural features are presented that lend support to the "ball-and chain" model, identify an interesting role for the c-terminal domain in gating, and establish the interaction pocket for AK-42. Electrophysiology and simulations nicely support the structural work. Overall, an elegant study, with high-quality data, and a well-presented manuscript.

Strengths

1. The cryoEM data presented reveals that the channel is in a closed conformation at depolarizing potential (0 mv). Structures for the closed state of CLCs were not previously available. A strong density for Glu205, which constitutes the Egate, allows for an unambiguous assignment of its position at the Scen Cl-binding site, thereby establishing the basis for the block in the closed channel.

2. The apo state particles were sorted into two classes that differ in the conformation of the CTD. A previously unobserved rearrangement of the CBS region in the CTD is reported wherein the CTD is positioned closer to the TM region in one of the subunits, breaking the C2 symmetry. The data implicates a role for the conformational flexibility of CTD in gating.

3. The most interesting finding of this work, is perhaps, the presence of an additional density, corresponding to a hairpin-like structure, that is seen only at the subunit where the CTD is positioned away from the TMD. The authors propose that the additional density corresponds to a 13 aa stretch in the N-terminal region. The position of the hairpin at the intracellular mouth of the CL-permeation pathway is likely to impede ion conduction, by a mechanism analogous to the "ball-and-chain" proposed in other voltage-gated channels.

4. The structure of CLC-2 in complex with a selective inhibitor AK-42 is in a conformation very similar to that of the apo state, with a clear additional density for the AK-42 molecule. Binding site interaction provides insights into AK-42 selectivity for CLC-2 vs CLC-1.

---

## [Author Response]

The following is the authors’ response to the original reviews.

**Reviewer #1 (Recommendations For The Authors):**
The discussion seems to imply that the ball-and-chain peptide is or is related to the common gate. (Although it isn't stated explicitly, it is implied based on the presentation of the gating model in Figure 8 immediately after the discussion of common gating, and the simultaneous opening of both pores in Figure 8). What does the asymmetric structure say about the relationship between the N-term peptide and common gating in ClC-2? It seems like this structure suggests that the CTDs can independently rotate, and independently bind N-terminal peptide, which might not be expected to impact both pores. Some additional clarification and/or discussion of these ideas could be helpful here.

We thank the reviewer for raising these very important points. We agree we should have been more explicit and have now expanded our discussion on this topic, highlighting the independent movement of the N-term peptide and CTDs and clarifying that it is currently unknown whether CLC-2 has a common gate (lines 431484).

Discussion of "Revised Framework for CLC-2 gating": I think this would be a little easier to follow if most of the legend from Figure 8 was in the main text at the end of that section. Also, additional labels in Figure 8 (of the glutamates, the N-terminal peptide, and what the CTD arrows represent).

We have revised this section of the text and added labels to the (revised) Figure as suggested.

Line 261: typo, misspelling of "hydrogen".

Fixed. (Now line 279.).

Figure 6 - supplement 2B: Looks like an error in numbering y-axis - should be 90/120/150, I think. Can you show the three data points for the WT initial current rectification? Can you clarify whether the 3 that you are analyzing are the ones where AK42 the AK42 "zero current" level is not more than the initial positive current?

We apologize for this error, which arose from the Y-axis label overlapping the tick labels, so 90/120/150 showed as 90/20/50. We have fixed this error and have added a new panel (C) to show three data points for the WT initial current rectification. In the Figure legend to panel C, we clarify that the 3 experiments we analyzed are the ones where the AK-42 current level is not more than the initial current at 80 mV.

**Reviewer #2 (Recommendations For The Authors):**
1. It appears from a close inspection of Figure 2 that the TM dimer is not quite symmetric, but I couldn't tell for sure from the figures as presented. No comment is made in the methods about symmetry imposed, and the authors explicitly comment on asymmetry in the cytoplasmic domain. It would be useful to have an explicit discussion of the TM dimer symmetry.

We have now explicitly stated that the TM dimer is symmetric, and we have clarified the wording in the Methods:

Main text, line 81: "The TM region of CLC-2 displays a typical CLC family symmetric homodimeric structure, with each subunit containing an independent Cl– pathway (Figure 2A, B)."

Methods (lines 557-558): "The following ab initio reconstruction and 3D refinement (for all structures presented in this paper) were performed with C1 symmetry (no symmetry imposed)."

1. For the simulations in Figure 5 Supplement 2, the N terminus flexibility is shown, but this of course can't be compared to a control. However, given the structural results, one might expect the JK helix to show changes in flexibility/mobility in the apo vs inactivated structures. Is this observed?

We agree that the structures strongly suggest the JK-helix is not as stable without the N-terminus bound. We did not perform comparative simulations on the JK helix in the apo vs inactivated structures. While we agree this could be of interest, we don’t think it is essential to our conclusions, and the simulations might need to be quite long to adequately capture dynamics of the JK helix. [In the simulation results shown in Figure 5 Supplement 2, our aim was to test the validity of the structure by determining whether the N-terminus remains bound to the channel in simulations. The plot shows that the N-terminus stays in the same binding pose with an average RMSD (to the initial structure) of less than 2 angstroms, which is generally considered to be relatively stable.].

1. I find the section "revised framework for ClC-2 gating" to be wanting. The ideas are illustrated in the cartoon, but should also be laid out in the text. In what ways are you revising the framework, and in what aspects are you carrying through ideas already proposed?

Thank you for raising this point, which was also raised by Reviewer 1. We have revised this section and the accompanying Figure (Figure 8 and Lines 431-484).

1. The authors mention in passing the idea that the hairpin could contribute to inward rectification (lines 227/8), but also suggest a role for the gating glutamate in this process. They also mention the idea of a common gate, but don't flesh out its function very much. These possibilities are very interesting and should be substantially fleshed out in the "framework" section, even if they cannot be fully answered yet.

We have expanded on these points in the “framework” section.

1. Figure 6E. points representing individual experiments should be shown.

We added points representing individual experiments for Delta N (normalized to WT) in the surface-expression experiments in Figure 6E. Individual data points for the electrophysiology experiments are in panel C; we did not replot these in panel E because some of the points would have been off scale.

1. The density in Figure 2A is hard to see, is there a better way to display it? Also, the orientation of the rightmost panel in Figure 2C is difficult to interpret.

We revised 2A to make the density easier to see. We revised Figure 2C so that the middle and rightmost panels have the same orientation.

1. P6. Line 87. This sentence is a little confusing, and perhaps could be a little clearer-the density is consistent with a Cl- ion, but no experiments have been done to support this, no?

We have clarified the wording as suggested (now line 89) and added references supporting Clˉ binding to the Sext site in CLCs (line 90).

1. P6 lines 89-98. Two lines of evidence, the conformation of the gate and the pinch point, both point to the structure representing a closed state. The wording as presented is a little hard to follow.

We have revised the wording in this paragraph (lines 92-111).

1. It's hard to distinguish water protons and oxygens in the lower right panel (QQQ).

We revised this panel (in Figure 3 – figure supplement 2) to better distinguish the water protons and oxygens.

**Reviewer #3 (Recommendations For The Authors):**
A few points to consider for improving the manuscript.1. It is intriguing that in the AK-42 structure, there is no density for the hairpin loop even though the CTD is in a symmetrical conformation as the apo. The authors could perhaps comment on whether there is any difference in the rectification properties of currents (or run-up) upon unblocking of AK-42 which may suggest that the hairpin binding is prevented by AK-42.

We have not yet performed the suggested experiment nor any experiments to examine state-dependence, though we agree such experiments would be informative. We have added a note on this point in the discussion, lines 334-337.

1. Although the conformation-dependent placement of the hairpin loop is convincing based on the density, the sequence assigned to this region is not conclusive.

To strengthen our conclusion concerning the hairpin assignment, we investigated fits of peptide segments from the disordered sections of the C-terminal cytoplasmic domain to the hairpin density. We found that these fits are not as good as that with the N-terminal peptide. This analysis is described in lines 179-181 and a new figure (Figure 5 – figure supplement 1). We appreciate the reviewer’s point that it is extremely difficult to conclusively assign residues that are not contiguous with the rest of the structure. Nevertheless, given the wide variety of evidence all pointing to the conclusion that the hairpin loop corresponds to residues 14-28, we think the assignment is on strong footing. We respectfully ask that you consider removing this criticism from the public review, as we think it will hinder the casual reader from recognizing the strength of the evidence: (1) of unresolved regions in CLC-2, residues 14-28 fit best; (2) residues 14-28 were previously identified as part of the ball blocking region (lines 158-161); (3) MD simulations support that the N-terminal residues stay stably bound (Figure 5 – figure supplement 4) (4) gain-of-function disease causing mutations map onto either the Nterminal residues or interacting residues on the TM domain (Figure 5 – figure supplement 6). Thank you for considering this request.

1. The authors should comment on the physiological relevance of the CBS domain rearrangements during gating.

We have added this sentence (lines 131-133): “The physiological relevance of C-terminal domain rearrangements is suggested by disease-causing mutations that alter channel gating (Estevez et al., 2004; Brenes et al., 2023).”

1. For the figures with cryo-EM maps, indicate the contour levels.

Contour levels are now indicated in the Figure legends.

1. It will be useful to the electrostatic map of the N-terminal peptide and the docking site.

This is now shown in Figure 5 – figure supplement 3 and Video 5.

1. Include a comment on the recent CLC-2 /AK-42 structure and if there are any differences in the structural features.

We added this text to lines 273-274: “The RMSD between our CLC2-TM-AK42 structure and that of Ma et al. is 0.655 Å, and the RMSD between the apo TM structures is 0.756 Å.”